



# Dynamic identification of snow phenology in the Northern Hemisphere

Le Wang[1], Xin Miao[1*], Xinyun Hu[1], Yizhuo Li[1], Bo Qiu[1], Jun Ge[1], Weidong Guo[1]

[1] School of Atmospheric Sciences, Nanjing University, Nanjing, China

*Correspondence to*: Xin Miao (miaoxin@nju.edu.cn)

**Abstract.** Snow phenology characterizes the cyclical changes in snow and has become an important indicator of climate change in recent decades. Changes in snow phenology can significantly impact climate and hydrological conditions. Previous studies commonly employed fixed threshold methods to extract snow phenology. However, these methods do not account for the variability in snow distribution across the Northern Hemisphere, leading to potential biases of snow phenology. In this

study, we observe that snow phenology extracted from different snow data and methods shows significant differences, but consistently underestimates snow duration at low and middle latitudes. Our analysis further indicates that the changes in snow depth exhibits a significant shift around 10% of peak value across the Northern Hemisphere, marking the transition between the snow and non-snow seasons. We further apply the 10% snow depth threshold and investigate the differences between original and newly extracted snow phenology. At low and middle latitudes, the snow cover duration (SCD) extends, the snow

cover onset day (SCOD) advances, and the snow cover end day (SCED) delays, especially on the Tibetan Plateau, where the SCD differences can reach 28 days. The change at higher latitudes is reversed. The dynamic snow phenology accounts for the spatial heterogeneity of Northern Hemisphere snow cover, and excludes the influence of inter-annual variability of snow cover on snow phenology extraction, providing a novel perspective for identifying and understanding snow cover variations in the Northern Hemisphere.

## 20 1 Introduction

Snow, an important component of the Earth's cryosphere, has also become a sensitive indicator of climate change (Brown and Mote, 2009; Dong, 2018; Kang et al., 2010). Approximately 98% of seasonal snow cover is concentrated in the Northern Hemisphere (Armstrong and Brodzik, 2001; Dietz et al., 2012a), and its variability has a significant influence on both the global climate system and the hydrological cycle (Déry and Brown, 2007; Cohen et al., 2012; Furtado et al., 2015; Harpold

and Brooks, 2018; You et al., 2020). Snow has high albedo and low thermal conductivity, regulating the surface energy balance and subsequently influencing atmospheric circulation (Marks et al., 1992; Gouttevin et al., 2012; Brutel-Vuilmet et al., 2013; Henderson et al., 2018). Meanwhile, the seasonal snow is an important natural reservoir that provides freshwater resources for more than a billion people (Barnett et al., 2005; Immerzeel et al., 2010; Sturm et al., 2017; Bormann et al., 2018). Therefore, accurate quantification of snow dynamics across the Northern Hemisphere (NH) is urgently needed.




Under global warming, the snow cover extent (SCE) of the NH has undergone a notable decline over the past few decades, with this trend projected to persist into the foreseeable future (Brown and Robinson, 2011; Estilow et al., 2015; Hori et al., 2017; Tang et al., 2022). Meanwhile, the variation rate of each season's average snow depth (SD) exhibited great fluctuations in different seasons. Compared with those in spring and autumn, the average SD in winter decreased at the highest rate (Xiao et al., 2020). In a warmer climate, the snow melting date will advance in time, and the melting amount will also increase.

(Barnett et al., 2005; Nijssen et al., 2001; Musselman et al., 2021). Moreover, the length of the snow season and the number of snow days are shortened (Notarnicola et al., 2022). At low and mid-latitudes, the continuity of snowfall becomes poorer, and snowfall time is more scattered and irregular (Li et al., 2022; Wang et al., 2024). These indicators suggest that snow in the Northern Hemisphere is undergoing significant changes.

        In recent decades, snow phenology has been widely used to characterize seasonal changes in snow. Common snow
phenology parameter indicators include snow cover onset day (SCOD), snow cover end day (SCED), and snow cover duration (SCD) (Liston, 1999; Liston and Hiemstra, 2011; Ke et al., 2016; Lin et al., 2017; Notarnicola, 2020). With climate change, snow phenology has changed significantly. Typically, SCD is shortened, SCOD is delayed, and SCED is advanced due to the increase in temperature (Whetton et al., 1996; Choi et al., 2010; Wang et al., 2013; Peng et al., 2013). However, there are regional differences in this variation. Opposite changes in snow phenology occur across northern middle and high latitudes

from 2001 to 2014, with SCD decreasing by 5.57±2.55 days at high latitudes and increasing by 9.74±2.58 days at mid-latitudes (Chen et al., 2015). Over the whole High Mountain Asia, the variability of snow phenology generally shows a decreased SCD, delayed SCOD, and advanced SCED, but the snow cover in the West Himalayas increases with increasing SCD (Tang et al., 2022). The SCD in China has exhibited an increasing trend of approximately 0.5 days per decade, while the trend is not significant on the Tibetan Plateau (TP) (Huang et al., 2020). Other results show that the spatial distribution of snow phenology

on TP is very uneven and exhibits temporal heterogeneity (Xu et al., 2023).

        Changes in snow phenology can significantly impact climate and hydrological conditions. Snow cover is a major contributor to runoff (Latif et al., 2020; Li et al., 2020). An earlier onset of snowmelt can modify the seasonal distribution of runoff, exacerbating the conflict between water supply and demand, and increasing the frequency of droughts and floods (Chen et al., 2016b; Wang et al., 2024). Changes in runoff dynamics driven by shifts in snow phenology may also affect vegetation

growth. Snow not only provides essential moisture for vegetation but also offers thermal insulation, protecting plants from harsh winds and cold temperatures. (Knowles et al. 2017; Liu et al. 2023). Wipf and Rixen (2010) have demonstrated that the timing of the vegetation growing season is notably affected by the advancement or delay of snowmelt. Therefore, thorough studies of spatiotemporal changes in snow phenology is essential for understanding regional and global climate dynamics, managing water resources, supporting vegetation growth, and predicting potential climatic crises.

Snow phenology was generally obtained through a two-step process in previous studies, i.e., identifying the presence or absence of snow in the grid based on a given threshold and calculating snow phenology indicators (Peng et al., 2013; Yang et al., 2019; Notarnicola, 2020). Various types of snow data are used to extract snow phenology, including SDs, snow cover fractions, and snow water equivalents, leading to possible differences in identified snow phenology (Chen et al., 2015; Guo et



al., 2022). Additionally, most studies have employed a fixed threshold to extract snow phenology in different regions and years
(Brown et al., 2007; Gao et al., 2011; Yue et al., 2022; Tang et al., 2022). The fixed threshold for snow phenology fails to
account for the variations in snow cover across the NH. In fact, snow cover increases with latitude, with thick and stable snow
cover at high latitudes and shallow and short-lived snow cover at middle and low latitudes, especially on the TP (Orsolini et
al., 2019). In addition, the snow changes from year to year due to many aspects of the climate. Snow conditions are variable,
but thresholds are always fixed, which can lead to underestimation or overestimation of snow phenology. Therefore, employing
different snow data and a fixed threshold will lead to uncertainties in extracted snow phenology. We aim to propose a new
method to extract the snow phenology which can reduce the errors caused by the fixed threshold method.

In this study, we compare the snow phenology extracted from different snow data and develop a dynamic threshold
method for snow phenology extraction across the NH. Section 2 describes the details of the data and snow phenology extraction
methods (including the traditional fixed threshold method and the new dynamic threshold method) used in this study. Snow
phenology extraction and comparison results are presented in Section 3. The discussion and conclusions are presented in
Section 4.

## 2 Data and Methods

### 2.1 Snow data

We use the Interactive Multi-Sensor Snow and Ice Mapping System (IMS; Helfrich et al., 2007) dataset to represent the change
in snow cover area (SCA). The IMS dataset is produced by the United States National Ice Center (USNIC) to provide cloud-
free snow cover products for the NH. The IMS dataset combines multiple optical and microwave sensors to classify snow and
non-snow areas. It offers three spatial resolutions of 1 km, 4 km, and 24 km, and the daily 24 km snow cover product covers
the period from early 1997 to the present, which is used in this work.

The 8-day snow cover component product (MOD10C2) is also used to represent the SCA variation. The MOD10C2
product delivered by the Moderate Resolution Imaging Spectrometer Satellite (MODIS) combines data from the 8-day
composite product of MOD10A2 at a resolution of 500 meters (Hall and Riggs, 2007). Utilizing an 8-day composite is
advantageous, as it accommodates areas where frequent cloud cover obstructs continuous surface observations (Frei et al.,
2012). It's worth noting that polar darkness prevents the mapping of snow cover in the arctic regions in boreal winter in this
dataset (Riggs et al., 2019). In the following this dataset will be abbreviated as SCF (snow cover fraction).
Meanwhile, we employ a separate dataset of snow depth (SD) for comparative analysis with the SCA dataset. The long-
term series of daily global SD are obtained by a passive microwave remote sensing inversion method (Che et al., 2019). The
remote sensing inversion method uses a dynamic brightness temperature gradient algorithm, which considers temporal and
spatial variations in snow characteristics and establishes a dynamic spatial and seasonal relationship between brightness
temperature differences at different frequencies and snow depth. Long time series of passive microwave brightness temperature



data are obtained from three sensors: SMMR, SSM/I and SSMI/S. The dataset, with a temporal resolution of 24 hours from 1980-2018 and a spatial resolution of 25067.53 meters, shows relative deviations within 30%.

The long-term series of daily global SD is affected by satellite orbits, leading to substantial missing measurements at low and middle latitudes. To minimize the negative effects of missing data, we replace the SD of the original dataset for the China regions with a long-term series of daily snow depth datasets in China (Che et al., 2015). This dataset is extracted from satellite-

borne passive microwave brightness temperature data using the Chinese passive microwave SD inversion algorithm of Che (Che et al., 2008). This data has been validated against meteorological observations, and absolute errors of less than 5 cm account for approximately 65% of all the data.

**2.2 Definition of Snow Phenology**

In this study, we calculate the snow phenology in the hydrological year, which is defined as the period from September 1 to

following August 31. For different data, snow cover phenology indicators are defined in different ways (Table 1). For the daily IMS binary SCE dataset, no additional processing is required to determine if the grid is covered with snow. However, a fixed threshold of snow cover is used to classify grids as snow-covered or snow-free for the SD datasets. Additionally, snow cover is considered present for the SCF dataset when the SCF exceeds 50% (Brown et al. 2007; Gao et al. 2011; Ke et al, 2016).

Following the identification of snow in the grid, SCD, SCOD, and SCED are extracted for every dataset. SCDs for the

SD and IMS datasets are calculated by summing snow-covered days. SCOD is defined as the first day with the first continuous snow cover exceeding five days in the hydrological year, whereas SCED is the last day with the last continuous snow cover exceeding five days. For the SCF dataset, considering the 8-day temporal resolution, the SCOD is the date four days before the first identified snow cover, and the SCED is four days after the last identified snow cover. SCD is determined by multiplying the number of snow occurrences by eight (Notarnicola, 2020; Yue et al, 2022; Guo et al, 2022).





**Table 1.** Definitions of snow phenological parameters for different datasets.

| Dataset | Threshold | SCD | SCOD | SCED |
|---|---|---|---|---|
| IMS | \ | $\sum_{i=1}^{n} Snow_i$ | the first day on which the pixel is first covered with snow for at least five of consecutive days | the last day on which the pixel is last covered with snow for at least five of consecutive days |
| SD | 2 cm | $\sum_{i=1}^{n} Snow_i$ | the first day on which the pixel is first covered with snow for at least five of consecutive days | the last day on which the pixel is last covered with snow for at least five of consecutive days |
| SCF | 50% | $\sum_{i=1}^{n} Snow_i * 8$ | the four day before which the pixel is first covered with snow | the four day after which the pixel is last covered with snow |

Note. Snow is the snow element used. n is the total number of days in the hydrologic year, which is 365 or 366 for the IMS and SD datasets and 46 for the SCF dataset.

**2.3 Dynamic Threshold for Snow Phenology**

In this study, we develop a dynamic snow phenology method with reference to the vegetation phenology extraction method. The vegetation phenology extraction method was proposed by White et al. (1997) from the normalized difference vegetation index (NDVI), which detects the start of growing season (SOS) and the end of growing season (EOS) across land cover. The formula is as follows:

$$NDVI_{radio} = \frac{NDVI - NDVI_{min}}{NDVI_{max} - NDVI_{min}}, \tag{1}$$

where $NDVI_{max}$ is the annual maximum NDVI, and $NDVI_{min}$ is the annual minimum NDVI. When the $NDVI_{radio}$ is above a certain threshold, the corresponding day of the year is determined as the SOS. When the $NDVI_{radio}$ exceeds is below a certain threshold, the corresponding day of the year is determined as the EOS. This approach enables the comparison of vegetation phenology across different land types, rather than using a fixed threshold (White et al., 2014; Yu et al., 2010; Sun et al., 2022).

To investigate the snow phenology in different areas across the globe, we propose a dynamic threshold for snow phenology.

$$Snow_{radio} = \frac{Snow - Snow_{min}}{Snow_{max} - Snow_{min}}, \tag{2}$$

where $Snow_{max}$ is the annual maximum snow index, and $Snow_{min}$ is the annual minimum snow index. The grid is determined to be snowy when $Snow_{radio}$ exceeds a certain threshold.





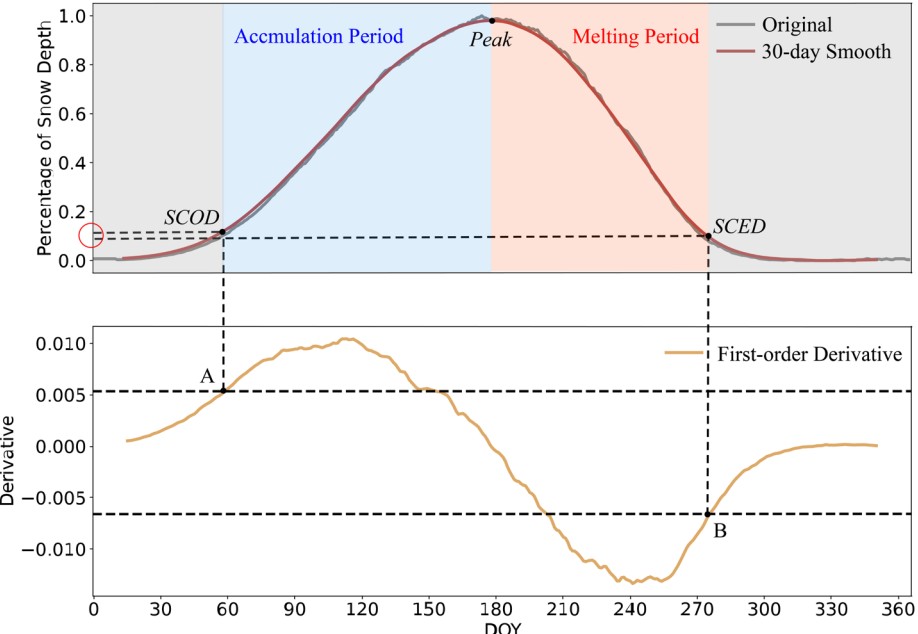

**Figure 1.** Intra-annual variability in the normalized SD in the Northern Hemisphere. The gray curves represent the original SD, the red curves represent the 30-day smoothed SD, and the yellow curves represent the first-order derivative. A and B represent the midpoints of extreme of the first-order derivative. Gray shading indicates the snow season is not start or is end, blue shading indicates the snow accumulation period, and red shading indicates the snow melting period.

To identify the optimal $Snow_{radio}$, we normalize and smooth the interannual SD variation curves for the entire NH and each latitudinal zone (including the Tibetan Plateau) with a 30-day moving window and then calculate their first-order derivatives. Smoothing is intended to remove noise. The first-order derivative represents the slope of the tangent line. Its extreme points show where the slope is steepest, meaning the curve changes most significantly. We assume the smoothed snow accumulation curve has a single peak structure. First-order derivative can be seen as the actual rate of snow accumulation or melting. The extreme points of the first-order derivative indicate maximum rate of snow changes. When the first-order derivative equals zero (beginning and end of the curve), it shows that snow has not started accumulating or has completely melted. The intermediate state between the maximum rate of snow changes and no change represents when snow starts to accumulate or melting is nearly complete, which is what we are looking for in SCOD and SCED. So, here we simply choose the extreme midpoint of the first-order derivative as the snow curve turning point. The percentage of turning point is the threshold we need. The above process is carried out for each grid in the NH, and Figure 1 shows a schematic of the entire NH. The midpoints of the extreme are labeled A and B, which correspond to the snow curves as SCOD and SCED. The percentages



for SCOD and SCED fall between 5% and 15% (marked by red circles). Below this threshold, the snow curve changes slowly, while above it, the curve changes rapidly. This threshold range (i.e. $Snow_{radio}$) can therefore serve as an indicator for the beginning/end of the snow season.

## 2.4 Elevation Data and Standard Deviation of Topography

To explore the relationship between topography and snow phenology, we use an elevation dataset with a spatial resolution of 0.008° derived from the NASA Digital Elevation Model (DEM). The dataset is used for gridded standard deviation of topography (SDtopo) calculations. Compared with the average topographic height, SDtopo provides a more accurate representation of topographic variability, which is essential for predicting snow cover distributions in mountainous regions
(Douville et al,1995; Swenson and Lawrence, 2012; Miao et al, 2022). Specifically, the 0.1° SDtopo data are obtained by calculating the standard deviation of all elevation values within each 0.1° grid cell.

## 3 Results

## 3.1 Comparison of Snow Phenology Extracted from Different Data

Due to the spatial heterogeneity of snow distribution in the NH, the diversity of snow data types and the choice of fixed
thresholds, the snow phenology extraction in the NH is subject to great uncertainty. To accurately capture snow phenology and reflect the effects of different methods on snow phenology, we first compare the snow phenology results of three datasets (IMS, SCF and SD) during the hydrological years 2000-2018. Notably, snow phenology above 60°N is not extracted from the SCF dataset due to the effects of the polar night.

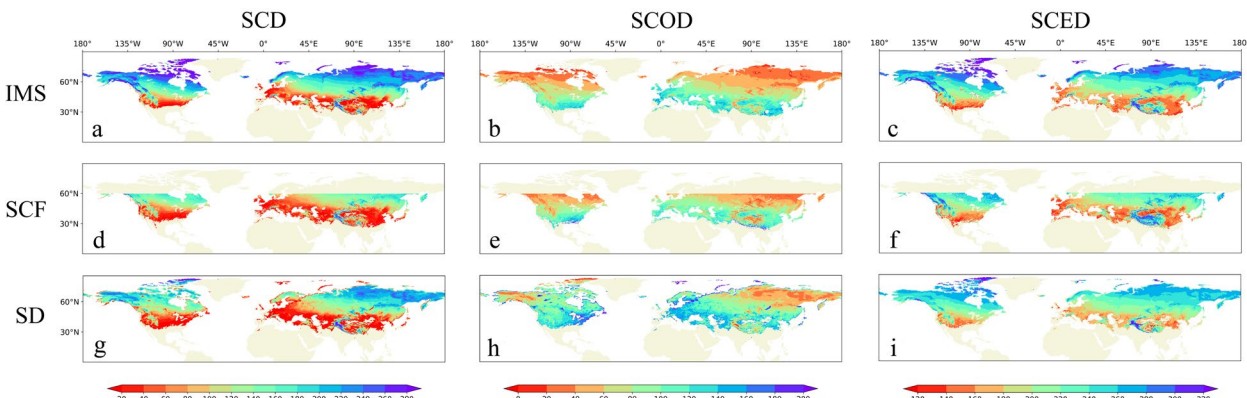


**Figure 2.** Spatial distribution of snow phenology extracted by the fixed threshold method over the Northern Hemisphere for the hydrological years 2000-2018. (a) Multiyear averaged snow cover duration (SCD) based on the IMS dataset. (b) Multiyear averaged snow cover onset day (SCOD) based on the IMS dataset. (c) Multiyear averaged snow cover end day (SCED) based on the IMS dataset. (d, e, f) Same as (a, b, c) but for the SCF dataset. (g, h, i) Same as (a, b, c) but for the SD dataset.




Overall, the spatial distribution of snow phenology across the NH exhibits pronounced heterogeneity, characterized by distinct latitudinal and altitudinal zonal patterns. In detail, as latitudes and altitudes increase, the SCD extends with earlier SCOD and later SCED (Fig. 2). At high latitudes (above 60°N), SCD generally exceeds 180 days, with SCOD occurring from September through October and SCED occurring from June through August. Within the latitudinal range of 40-60°N, SCD exhibits a broader range from a minimum of approximately 20 days to a maximum of around 180 days. SCOD occurs from November to December, while SCED occurs from March to May. In areas below 40°N, except for highlands on the TP, SCD is generally less than 30 days. snow phenology on the TP increases with altitude, with SCD being greater than 280 days in the western mountainous region, but less than 20 days in the central basin region.

In the NH, snow characteristics across various latitudinal zones exhibit notable distinctions, leading to diverse snow phenology patterns. To thoroughly assess the impact of varying snow data types and extraction methods on the snow phenology results, we further compare snow phenology across different latitudinal zones (including the Tibetan Plateau) using three datasets. From Table 2, the statistical results reveal a substantial difference (22 days) in SCD between the IMS and SD datasets across the entire NH. Specifically, this difference is primarily attributed to SCOD, which shows a notable variation of 33 days, while SCED displays a marginal difference of only one day.


**Table 2.** Different snow phenology indicators extracted from the three datasets in different latitudinal zones (including the Tibetan Plateau).

| snow phenology parameters | dataset | NH | TP | 30-40°N | 40-50°N | 50-60°N | 60-75°N |
|---|---|---|---|---|---|---|---|
| | IMS | 110 | 89 | 30 | 86 | 163 | 229 |
| SCD | SCF | \ | 93 | 46 | 81 | 150 | / |
| | SD | 132 | 97 | 39 | 51 | 120 | 176 |
| | IMS | 66 | 86 | 109 | 91 | 64 | 37 |
| SCOD | SCF | \ | 70 | 103 | 80 | 51 | / |
| | SD | 99 | 88 | 121 | 125 | 100 | 77 |
| | IMS | 226 | 224 | 168 | 190 | 232 | 268 |
| SCED | SCF | \ | 245 | 175 | 189 | 229 | / |
| | SD | 227 | 211 | 182 | 189 | 225 | 257 |

Note. NH is short of the Northern Hemisphere and TP is short of the Tibetan Plateau.

Further analysis at different latitudinal zones emphasizes the significant variability in snow phenology among the three datasets. For the SCD, the differences among the three datasets are relatively small in the middle- and low-latitude zones and




are more pronounced at high latitudes (53 days) between IMS and SD. Besides, the SD dataset consistently identifies the latest SCOD across all latitudinal zones, while the SCF dataset identifies the earliest SCOD, with the difference between them exceeding 1 month north of 40°N latitude. In comparison to the above two snow phenology indicators, the SCED has smaller differences, exhibiting pronounced magnitudes only at low and mid-latitudes. The most substantial deviation is observed

between the SCF and SD datasets on the TP, reaching 34 days.

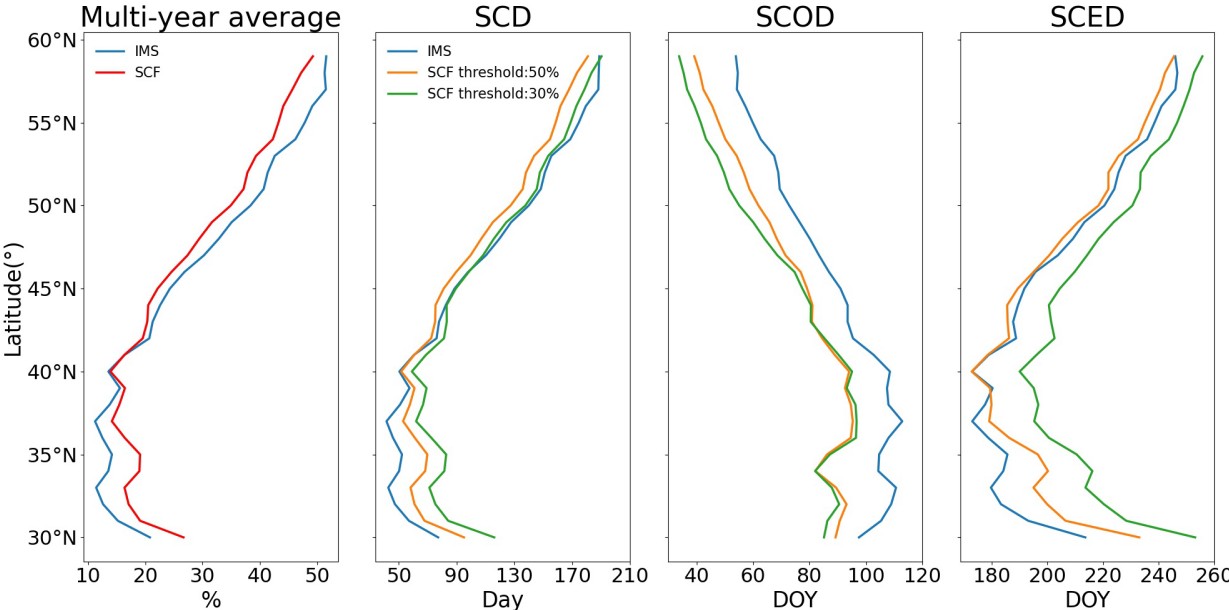

**Figure 3.** Changes in snow phenology indicators from the IMS and SCF datasets with latitude. The figures indicate the latitudinal variations of (a) the snow cover fraction (SCF), (b) snow cover duration (SCD), (c) snow cover onset day (SCOD), (d) snow cover end day (SCED).

The blue curve in (a) represents IMS, and the red curve represents the SCF. The blue curves in (b, c, d) represent the snow phenology extracted by IMS, the orange curves represent the snow phenology extracted by SCF with a threshold value of 50%, and the green curves represent the snow phenology extracted by SCF with a threshold value of 30%.

Among the three datasets, IMS and SCF both serve as representations of snow coverage. The comparative analysis

indicates that both datasets exhibit synchronous latitudinal variations, with a turning point at 40°N (Fig. 3). South of 40°N, the mean snow coverage from IMS is larger than that of SCF, and the opposite is true for the north of 40°N. To explore the influence of threshold selection on snow phenology extraction, different thresholds of 30% and 50% are also implemented within the SCF dataset. The results show that the variations of snow phenology extracted from the two datasets are basically the same. For SCD, 40°N also serves as a turning point, with IMS extracting the shortest SCD south of 40°N and the longest

SCD north of 40°N. Additionally, IMS consistently extracts the latest SCOD across all latitudes, while SCED is the latest for SCF with a 30% threshold. There are significant differences in snow phenology when different thresholds are used. Specifically,





when the threshold is reduced, conditions with snow grids are more easily reached, resulting in a longer SCD, earlier SCOD, and later SCED. This indicates the importance of threshold selection in accurately capturing snow dynamics and temporal changes in snow coverage.


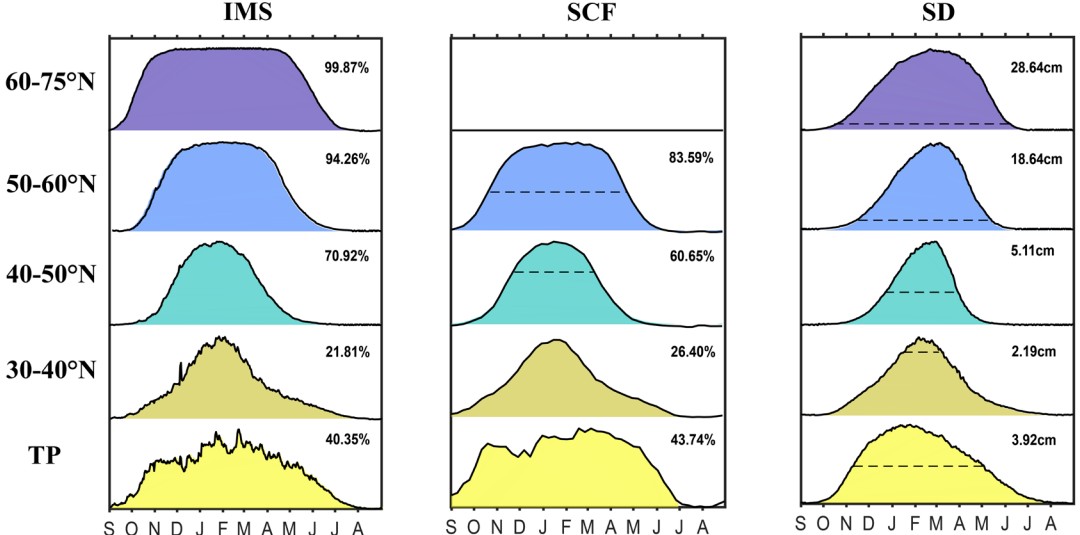

**Figure 4.** Intra-annual variations of IMS, SCF, and SD in five latitudinal zones (including the Tibetan Plateau). The dashed lines in SCF curve represent the SCF of 50%, and the dashed lines in SD curves represent the SD of 2 cm. The SCF dataset north of 60°N is not analyzed due to the effects of the polar night. The values in the graphs characterize the annual maximums of different snow elements.


Temporally, the IMS, SCF and SD datasets show different intra-annual variations (Fig. 4). At low and middle latitudes, such as the latitudinal zone of 30-40° N, the annual average maximum SCF is 26.4%, and the maximum SD reaches 2.19 cm. Although perennial snow is prevalent at high elevations on the TP, the annual mean maximum SCF remains below 50%, and the maximum SD is only 3.92 cm. It is evident that the thresholds of 50% for SCF and 2 cm cannot realistically characterize
the seasonal variations of snow at low and middle latitudes.

Furthermore, SD consistently displays a stable, single-peak change, effectively capturing the snow processes from accumulation to ablation. In contrast, the IMS and SCF curves exhibit significant fluctuations at low latitudes, particularly on the TP, where multiple peaks are observed. As latitude increases, the IMS and SCF curves become smooth. However, in the latitudinal zones north of 50°N, snow cover can become extensive or even reach complete coverage over time, masking the
variability of snow and resulting in a lack of distinct peaks. In our study, snow peak is also used as an indicator, and snow accumulation and melting processes are analyzed separately. The stable single-peak structure of SD curve is more suitable for our study, so the subsequent snow phenological correction is to use SD data.



Based on the above results, we believe that the characterization of snow phenology is significantly influenced by the selection of snow data and the methods used for extraction. In the next section, we aim to enhance the snow phenology
extraction method using SD data to obtain more reasonable snow phenology in the NH.

**3.2 Dynamic Snow Phenology Threshold**

Snow in the NH shows similar latitudinal distribution as vegetation, while vegetation decreases with increasing latitude (Wang et al. 2016; Zeng et al, 2020), and snow shows the opposite trend. One of the commonly used methods for extracting vegetation phenology is the dynamic threshold method (see Section 2.3). In this approach, grid points have varying thresholds based on
vegetation status, enabling the effective capture of vegetation dynamics by comparing each pixel's phenology with its seasonal changes. Given that the zonal variations of the NH snow cover are similar to those of vegetation, we suggest that a dynamic threshold method could better capture the spatial characteristics of snow phenology compared to a fixed threshold. Therefore, we propose a dynamic threshold method for snow in the NH.  To improve the accuracy of the analysis, we extend the time period to the hydrological years 1989-2018.


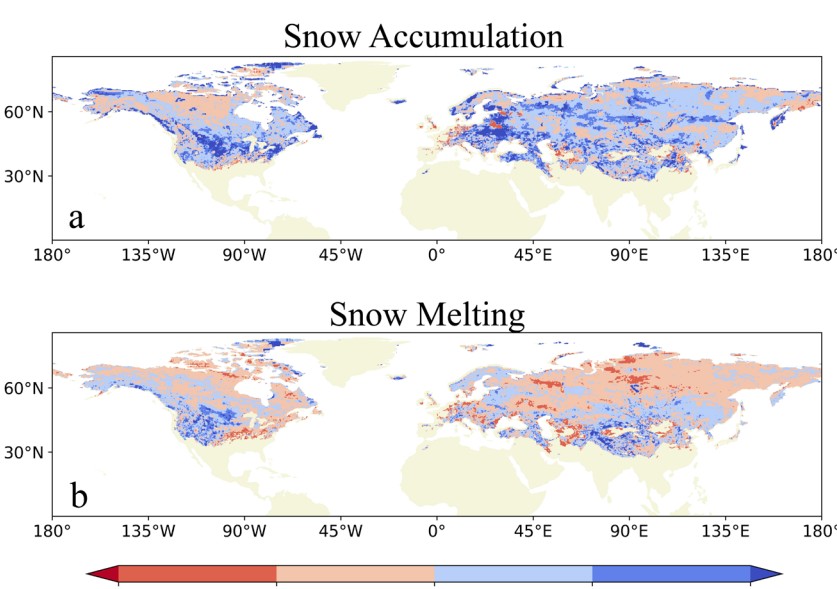

**Figure 5.** Spatial distribution of Northern Hemisphere threshold percentages extracted from the first-order derivatives. (a) Percentage thresholds associated with snow accumulation (SCOD) from the first half of the first-order derivative maximum value. (b) Percentage thresholds associated with snow melting (SCED) from the last point of half of the first-order derivative minimum value. The extraction of
threshold percentages is preceded by a sliding average process with a window of 30%.



By calculating first-order derivatives over five latitudinal zones in the NH to determine $Snow_{radio}$, we find that the $Snow_{radio}$ values in these bands almost fall within the range of 5% to 15%. Conducting the same analysis on each grid point across the entire NH further validates the generalizability of this percentage threshold interval. The results show that 73.05% of the areas during the snow accumulation period and 82.65% of the areas during the snow melting period fall within the 5-15% interval (Fig. 5), which suggests that this percentage range can be applied to the entire NH. Notably, here we calculate the area percentage after masking the area with SCD less than 10 day. Based on these findings, 10% threshold (the midpoint between 5% and 15%) is established for determining the presence of snow in a grid. The grid is considered snow-covered if its snow depth reaches a value corresponding to 10% of its maximum value minus its minimum value; otherwise, the grid is considered to be snow-free.

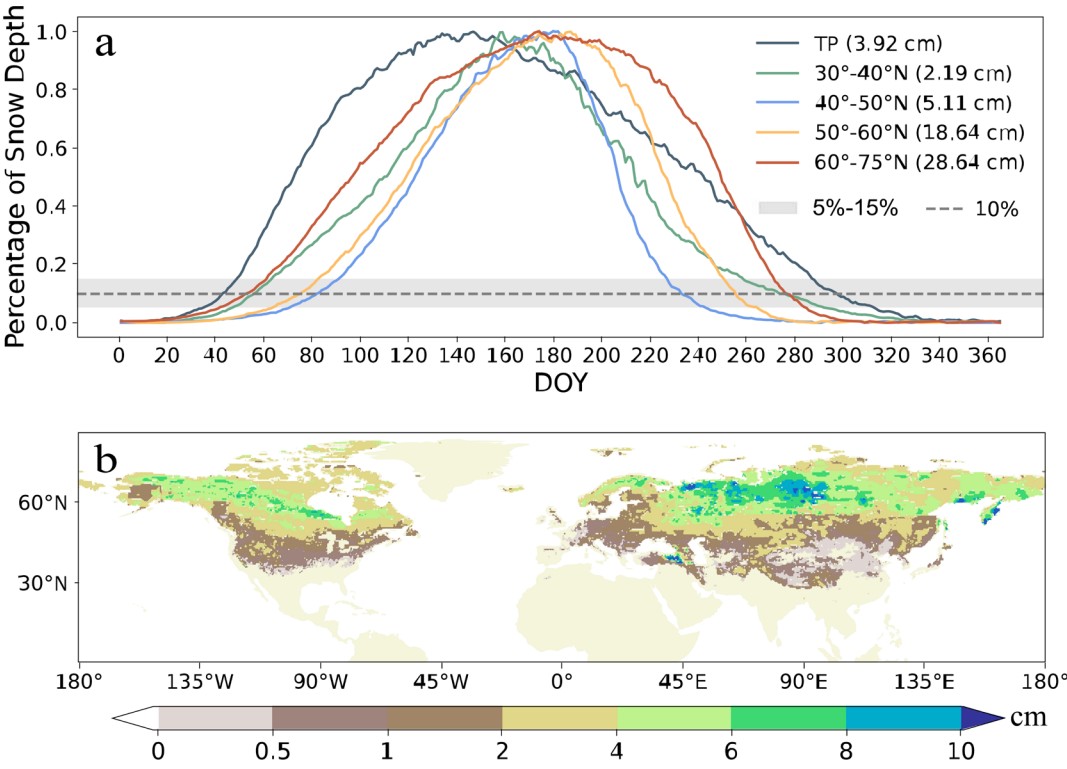

**Figure 6.** The dynamic SD threshold across the Northern Hemisphere. (a) Intra-annual variability in the normalized SD for five latitudinal zones. Shading represents the interval 5%-15%, and the dashed line represents the dynamic threshold of 10%. Actual maximum snow depths for each latitude band are in parentheses. (b) Spatial distribution of multi-year average snow thresholds in the Northern Hemisphere extracted using the snow dynamics threshold method.





Normalizing and plotting the intra-annual variation of SD curves for five latitudinal zones on the same graph, it is evident that the 10% threshold generally corresponds to the position of the turn in the curve slope for each latitudinal zone (Fig. 6a).

The percentage SD curves show minimal change below 10% threshold, indicating little snow cover and that the snow season has either not started or has ended. In contrast, the curves rise/fall sharply when percentage SD curves are above 10% threshold, indicating ongoing snow accumulation/melting. The position of 10% is exactly at the inflection point of the transition between the two states of snow, which can be recognized as an indicator for judging the beginning/end of the snow season. Meanwhile, the unified results of multiple latitudinal bands confirm the universality and reasonability of 10% threshold.

The SD curves exhibit a similar single-peak pattern across latitudes, with the curves at high latitudes (60-75°N) displaying the widest shape, gradually narrowing towards lower latitudes. This trend is consistent with the gradual reduction in SCD with decreasing latitude. An exception is observed in the SD curve of TP, which shows a broad shape similar to that of high latitudes, even with a leftward shift in its position. This suggests that TP has a longer SCD and earlier SCOD than expected, despite being located at low to mid-latitudes. The SD's peak also increases with latitude. North of 60°N, the SD's peak reaches 28.64

cm, while south of 40°N, it is only 2.19 cm. In the TP region, the peak snow depth is 3.92 cm, occurring approximately one month earlier compared to other latitudinal zones. The timing of snow peaks plays a crucial role in influencing the dynamics of snow accumulation and melting processes.

Employing the 10% dynamics threshold method, we extract SD thresholds for 30 years in the NH and then average (Fig. 6b). The cutoff for the traditional 2 cm threshold falls within the mid-latitudes, approximately between 40°N and 50°N. At

low- and mid-latitudes, the thresholds tend to be less than 2 cm and even below 0.5 cm on half of the TP. In contrast, high-latitude SD thresholds exceed 2 cm. The majority of the extracted snow depth thresholds at high latitudes range from 4 to 6 cm. These findings indicate that employing a fixed SD threshold of 2 cm overestimates the snow phenology for high latitudes and underestimates it for regions at low- and mid-latitudes. Each grid in the Northern Hemisphere has varying SD thresholds annually, and the multi-year average SD threshold pattern closely resembles the spatial distribution of SD. This suggests that

the method can dynamically adjust the threshold based on the annual and regional SD's variations, thereby reducing the errors in snow phenology extraction caused by large-scale climatic influences.



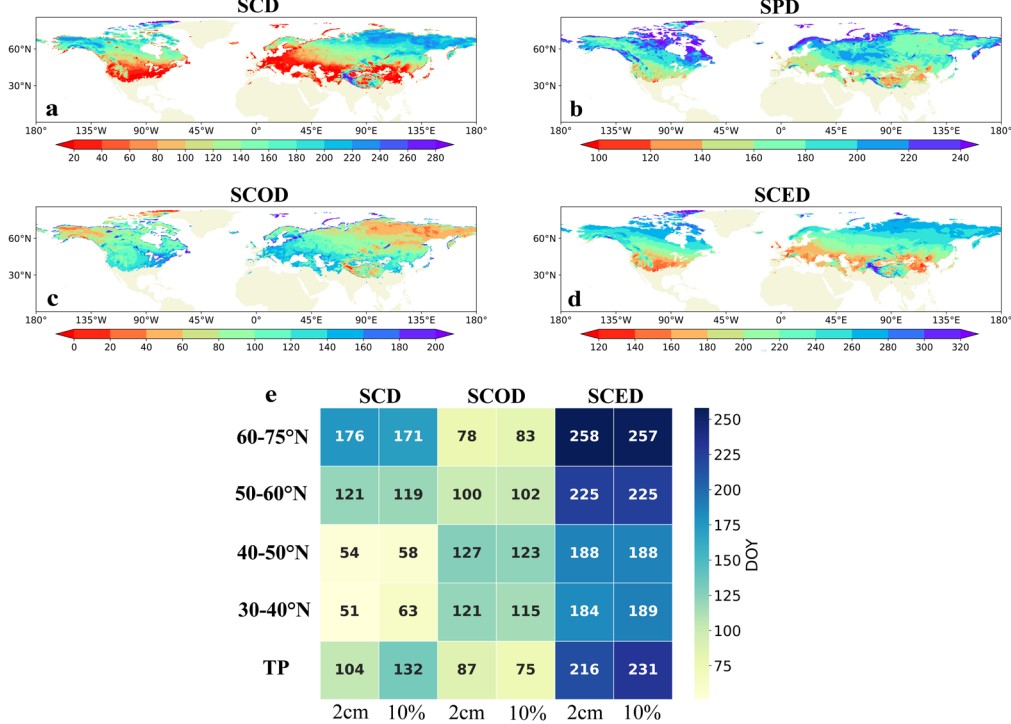

**Figure 7.** Spatial distributions of (a) snow cover duration (SCD), (b) snow peak days (SPD), (c) snow cover onset day (SCOD), and (d) snow cover end day (SCED) in the Northern Hemisphere extracted using the dynamics threshold method. (e) Hotspot map of comparison in snow phenology extracted by the snow dynamic threshold method and the traditional fixed threshold method in five latitudinal zones.

The temporal and spatial distributions of snow cover vary across different regions, leading to varying snow peak times. The time of snow peaks is crucial for the hydrological and ecological dynamics of climate systems and should be included as an indicator of snow phenology. Here, we propose a novel snow phenology index named the snow peak day (SPD). Next, we extract the SCD, SCOD, SCED and SPD using the dynamic SD threshold during the hydrological years 1989-2018. The spatial distributions of the snow phenology indicators are similar to those of the original method with increasing latitude and elevation, SCD lengthens, SCOD advances, and SCED delays. Furthermore, SPD also exhibits distinct latitudinal and altitudinal characteristics, with SPD typically occurring in January at mid- and low- latitudes, in April at high latitudes, and even later in perennial snow regions. It is noteworthy that the SCD and SPD exhibit contrasting patterns in Europe and North Asia in around 60°N latitude. While Europe experiences a shorter SCD than North Asia does, the SPD occurs later, implying sustainable snow accumulation at the onset of the snow season. In contrast, North Asia displays a longer SCD but with less accumulated snow, resulting in reduced snow cover towards the later period and an earlier SPD. These findings emphasize the unique significance





of each snow indicator, highlighting their complementary nature. Therefore, analyzing multiple snow phenology indicators

can help to comprehensively understand the snow evolution process from multiple perspectives.

Here, we further compare the results extracted using the traditional fixed threshold method and the new dynamic threshold method. Overall, the SCD and SCOD are significantly different between the two methods, but SCED varies relatively little. This suggests that snowmelt is a rapid and transient process that is less sensitive to the choice of threshold. Besides, it can still be seen that 40-50°N is a dividing line (Fig. 7e), where there are no significant differences between the two methods. In areas

north of 50°N, employing the dynamic threshold method leads to a reduction in SCD and a retardation in SCOD compared with those of the traditional method, while the modification in SCED proves non-significant. Conversely, at low- and mid-latitudes below 40°N, SCD is extended, SCOD is advanced, and SCED is delayed. Thus, the reduced SD threshold induces a longer snow phenology at low- and mid-latitudes. Noteworthy, the TP shows the most marked divergence among five regions, with SCD increasing by 28 days, SCOD advancing by 12 days, and SCED delaying by 15 days. The application of snow

dynamic thresholds yields the most substantial difference for TP in comparison to fixed thresholds.

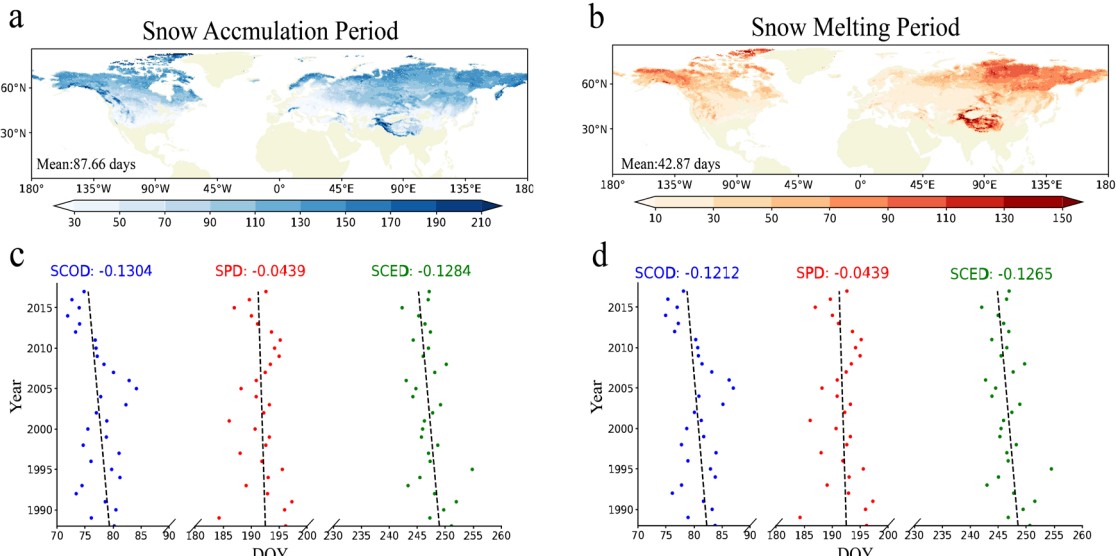

**Figure 8.** Spatial distribution of snow depth during the (a) snow accumulation period and (b) snow melting period in the Northern Hemisphere extracted using the snow dynamics threshold method. Scatterplots of SCOD, SPD and SCED extracted using (c) the traditional

2 cm fixed threshold method and (d) the snow dynamics threshold method in the Northern Hemisphere. The blue dots represent SCOD, the red dots represent SPD, and the green dots represent SCED. The numbers are the slopes of the linearly fitting lines.

We further investigate the spatial distribution and temporal variations of snow season. The interval between SCOD and SPD is classified as the snow accumulation period, while the duration from SPD to SCED is the snow melting period. Across





the NH, the average length of the snow accumulation period significantly exceeds that of the melting period, with mean durations of 88 days and 43 days, respectively. Notably, both the snow accumulation and melting periods are longer with increasing latitude and altitude. Over the last three decades, the NH's SPD has advanced by -0.0439 days per year, indicating an advancement in the SPD over time. Employing the snow dynamic threshold method, we observe an increase in the trend of the SCOD from -0.1304 to -0.1212 days per year, while the trend of the SCED increases from -0.1284 to -0.1265 days per year. These changes suggest a lengthening of the snow accumulation period and a shortening of the melting period as time progresses. However, the magnitude of the extended accumulation period has diminished, and the reduction in the magnitude of the melting period is also notable.

In summary, the interannual variability and heterogeneity in spatial distribution of snow highlight the limitations of using fixed thresholds at different times and regions. Therefore, it is unreasonable to use a uniform SD threshold of 2 cm as a criterion for assessing snow phenology across the entire NH. Through the implementation of a snow dynamic threshold method, the snow phenology of individual pixels can be evaluated in relation to their unique seasonal fluctuations, allowing for a more accurate representation of the actual snow phenology. In addition, the snow phenology dynamic threshold method is more reasonable in areas with complex topographic and climatic features, such as the TP.

### 3.3 Characteristics of TP Snow Phenology

In the previous sections, we elucidate that the TP exhibits the most substantial alterations in snow phenology after using the novel method. Consequently, this region is the focus of our attention in the following analysis.

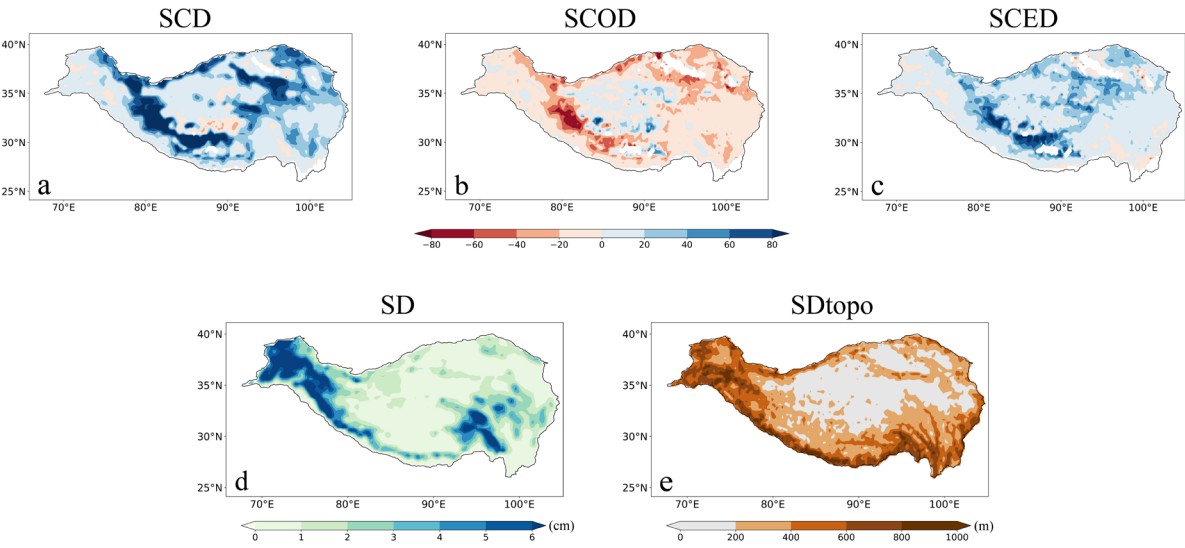



**Figure 9.** Spatial distribution of differences in (a) snow cover duration (SCD), (b) snow cover onset days (SCOD), and (c) snow cover end
day (SCED) extracted by the snow dynamic threshold method and the traditional fixed threshold method on the Tibetan Plateau. Spatial
distribution of (d) multi-year mean snow depth (SD) and (e) standard deviation of topography (SDtopo) on the Tibetan Plateau (TP).

Shallow and unstable snow covers the central TP with a short SCD of rarely more than one month. Conversely, the SCD
of perennial snow cover in the mountainous areas of the western TP can exceed 10 months. Meanwhile, SCD can reach
approximately 200 days in the southeastern TP due to sufficient water vapor, with SCOD typically occurring in October and
SCED occurring in May. Figure 9 demonstrates that the dynamic threshold approach induces extended SCD, advanced SCOD,
and delayed SCED. The largest difference in the snow phenology is concentrated in the southwestern and east-central TP, with
disparities of up to 165 days in SCD, 126 days in SCOD, and 113 days in SCED. Furthermore, there are consistent trends of
SCD and SCED across the TP, while SCOD shows a certain variability. There is an advanced SCOD in the majority of the TP,
particularly in high altitude areas, while a slight delay in SCOD is observed in the central TP.

Our analysis indicates that the spatial distribution of snow phenology on the TP and the differences between the two
methods are closely related to the topography. The TP has a diverse topography, with SDtopo below 200 m in the central
region and over 200 m and even over 800 m in the mountainous areas of the northwestern and southeastern TP (Fig. 9e). The
complex topography causes spatially heterogeneous land surface characteristics and snow conditions (Helbig et al., 2015).
Obviously, the SD on the TP clearly shows a spatial pattern that is consistent with that of the SDtopo. SD exceeds 10 cm in
the northwestern and southeastern TP, whereas it is less than 2 cm in the central region. Therefore, we divide snow on the TP
into two categories based on Sdtopo: mountain snow areas and non-mountain snow areas. Areas where SDtopo exceeds 200
m are classified as mountain areas, while those with SDtopo values below 200 m are considered non-mountain areas.





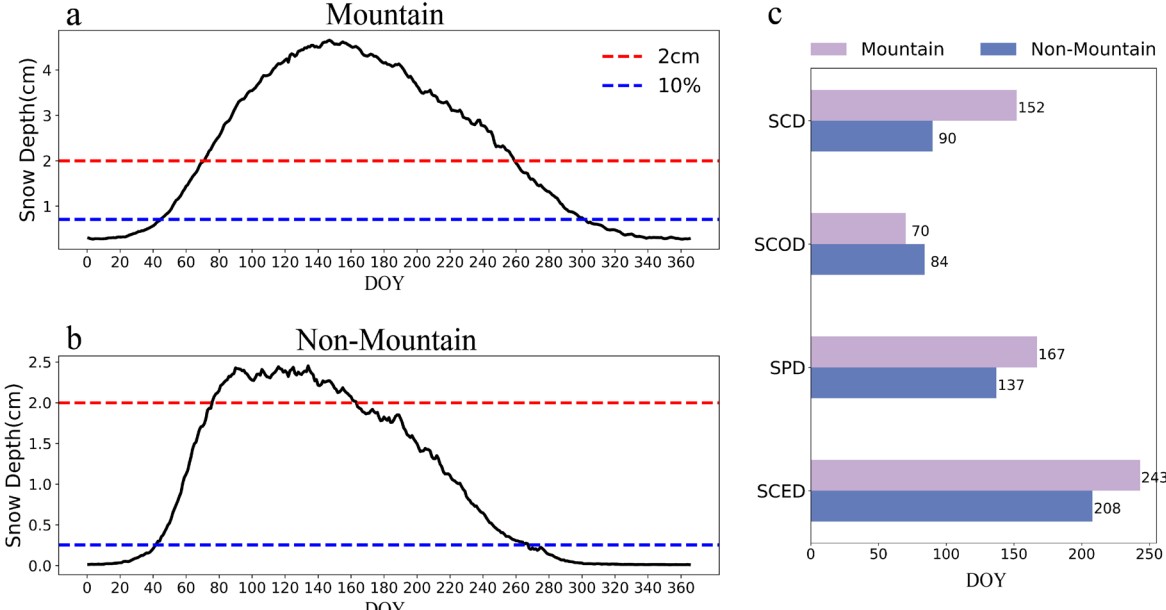

**Figure 10.** Intra-annual variation of snow depth in (a) mountain, and (b) non-mountain areas on the Tibetan Plateau (TP). The red dashed line represents the location of the fixed threshold of 2 cm, and the blue line represents the location of the threshold extracted using the snow dynamic threshold method. (c) Histogram of snow phenology in non-mountain and mountain regions of the TP using the dynamic threshold method.

The snow phenology characteristics significantly differ between mountain and non-mountain areas of the TP. Specifically, the SCD in mountain areas amounts to 152 days, whereas it is significantly shorter in non-mountain areas (90 days). The SCD's variation is also evident across different snow phenology indicators. The SCOD occurs earlier in mountain areas, and the SPD and SCED are later. This discrepancy is attributed to the higher altitudes and lower temperatures observed in mountain regions, leading to more snow and then delayed snow peaks and SCED. In contrast to the NH overall, the difference in SCED between the TP mountains and non-mountainous areas is more significant than that of SCOD, which plays a greater role in shaping the SCD differences. The discrepancy in SCOD between these two areas is 14 days, whereas the discrepancy in SCED extends to 35 days.

In mountain areas, the annual average maximum SD reaches 4.653 cm. Employing the dynamic snow threshold method led to a reduction from the original 2 cm threshold to 0.712 cm, which is more consistent with the position of the turn change in the curve slope. Conversely, snow in non-mountain areas remains shallow and unstable, with an annual average maximum SD below 2.5 cm. After using the dynamic threshold method, the threshold is adjusted to 0.254 cm. The original 2 cm threshold nearly reaches the snow peak, indicating a great underestimation of snow phenology. However, a 0.3 cm threshold may be excessively low and too easy to reach, potentially losing its phenological significance. For areas with shallow snow, the snow cover fraction (SCF) may capture more accurate snow phenology information than the SD. This is because the SD at individual



grid points within sparse snow cover fails to accumulate cumulatively, whereas the SCF effectively captures the transition from less to greater snow cover. Therefore, the use of SCF dataset may be more helpful in extracting accurate snow phenology information when analyzing shallow and unstable snow areas.

The distribution of TP snow cover has significant spatial heterogeneity due to the complex topography. Consequently, snow phenology is also profoundly influenced by topographic factors. Specifically, with increasing SDtopo, the SD becomes larger, SCD becomes longer, SCOD advances, and SPD and SCED delay, which leads to distinctive differences in snow phenology between mountain and non-mountain areas. The snow dynamic threshold method demonstrates substantial enhancements for snow phenology in mountain areas. However, the availability of this method in improving snow phenology assessments in non-mountain areas with shallow snow requires further investigation.

## 4 Conclusions and Discussion

In this study, we explore the spatial distribution of snow phenology in the Northern Hemisphere (NH) using several sets of satellite remote sensing snow data and multiple methods. A new extraction method for snow phenology is proposed in the NH, and the differences in snow phenology using the two methods are compared to evaluate new snow phenology method. The conclusions are described below.

Snow phenology extracted from the fixed threshold method and different snow data (SD, SCF, and IMS) exhibit approximately the same spatial distribution across the NH. As the latitude and altitude increase, SCD extends, SCOD advances, and SCED delays. However, notable discrepancies exist in the SCD and SCOD across the datasets, with peak variances reaching 53 and 49 days, respectively. In contrast, the SCED exhibits less variability, with a maximum difference of merely 34 days.

Considering the inter-annual variability of snow as well as regional differences in the NH, the dynamic method is more applicable to the extraction of snow phenology.The 10% threshold coincides with the inflection point of the rapid change in the snow depth curve, which marks the entry into the snow season. The new method induces a shortening of SCD and a delay in SCOD at higher latitudes, while SCED exhibits minimal change. In contrast, at lower latitudes, the adjustments in these metrics are inversely related.

The snow phenology experiences the most substantial changes on the TP when the new method is used, with SCD increasing by 28 days, SCOD advancing by 12 days, and SCED delaying by 15 days. Due to complex topographic and climatic features, there are also large differences in snow between mountainous and non-mountainous areas on the TP. The mountainous areas have longer SCD, earlier SCOD and later SPD and SCED , with a substantial 62-day variation in SCD. Meanwhile, The dynamic threshold method for snow phenology is well-suited for analyzing mountainous regions of the TP. However, its applicability to non-mountainous areas with shallow snow remains to be further investigated.

This study proposes an algorithm based on dynamic thresholds to recognize snow phenology, similar to vegetation phenology. The snow accumulation period corresponds to the start of season (SOS), and the melting period aligns with end of



season (EOS). Similar to vegetation, the state of snow in the NH shows different trends (Armstrong et al., 2001; Brown et al., 2011; Guo et al., 2021). Compared to the fixed threshold method, the dynamic method improves accuracy by adjusting

thresholds according to regional characteristics and temporal changes (Burgan and Hartford, 1993; White et al., 1997; White et al., 2006). It selects inflection points, peaks, or specific percentiles on snow curves to be more flexible and universal. Therefore, the approach is particularly suitable for large-scale and multi-temporal studies of snow phenology and demonstrates significant advantages when dealing with complex environmental changes. By dynamically adjusting snow thresholds, each year's accumulation and melting periods are determined based on actual conditions, replacing the dependence on traditional

seasonal divisions. This offers a new 'reference system' for describing seasonal changes, similar to the Twenty-four Solar Terms in climate research (Qian et al., 2010, 2012), providing a novel perspective and method for understanding snow changes in the Northern Hemisphere under global warming.

       Despite improvements in snow phenology extraction, variations in the data and definitions of snow phenology and hydrological years lead to differences in extracted snow phenological characteristics, which are further compounded by

inherent data uncertainties (Xie et al., 2017; Ma et al., 2020; Guo et al., 2022). The fundamental principles underlying snow information acquisition vary across observation methods, impacting binary snow results (Hall and Riggs, 2007; Dietz et al., 2011; Zhang et al., 2024). Factors like the observational instrument's orbit and cloud cover can further affect the accuracy of snow datasets (2005; Gao et al., 2010; Coll andLi, 2018). Second, the performance of snow data varies geographically. For instance, snow depth data is more effective for analyzing snow phenology in regions with consistent snow cover, while in

areas with shallow snow, it may not accurately capture accumulation and melting processes. In addition, for the transient snow area, the snow depth curve is more volatile, which makes the assumed single-peak structure untenable. After comprehensive consideration, the snow cover area may be a more reliable indicator in such cases. These factors heighten uncertainty in snow phenology analyses, emphasizing the need for careful dataset selection and methods. Therefore, it is necessary to integrate ground observation data with different remote sensing data to form a more comprehensive and accurate snow phenology

extraction system.

       Furthermore, we propose a dynamic approach to defining snow phenology by adjusting the threshold for snow presence in this study. Notably, snow phenology extraction involves two key steps: identifying snow presence and applying a consecutive snow days threshold to extract the snow phenology. These consecutive days can mitigate the influence of sporadic snowfall events. While we retain the traditional five consecutive day threshold, it may not be suitable for all regions. In some

areas, such as the central TP, shallow snow plays an important role whose surface albedo has a strong influence on snow ablation (Wang et al., 2020). Moreover, snow there is highly discontinuous over time, with an annual average of 14 snow cover events and prolonged periods without snow cover (Li et al., 2022; Wang et al., 2024). It is difficult to achieve five consecutive snow days, leading to barriers to identifying the snow season. Future work will take the dynamic consecutive snow days threshold into account.

In conclusion, this study reveals significant differences in snow phenology extracted from diverse snow data and methods and reveals that employing the fixed threshold method cannot accurately capture the actual snow season. We therefore develop





a novel physically based snow phenology extraction method based on a spatiotemporally dynamic threshold, enhancing the snow phenology extraction in the Northern Hemisphere, especially on the TP. The dynamic snow phenology accounts for the spatial heterogeneity of Northern Hemisphere snow cover, and excludes the influence of inter-annual variability of snow cover

on snow phenology extraction, providing a novel perspective for identifying and understanding snow cover variations in the Northern Hemisphere.



*Competing interests.* The contact author has declared that none of the authors has any competing interests.

*Acknowledgments.* This study is jointly supported by the Second Tibetan Plateau Scientific Expedition and Research Program (STEP) (2019QZKK0103), and the National Natural Science Foundation of China (42305033 & 42375115).

*Data availability.* The Interactive Multi-Sensor Snow and Ice Mapping System (IMS) is downloaded at https://nsidc.org/data/g02156/versions/1/. The 8-day snow cover component product (MOD10C2) is available at
https://nsidc.org/data/mod10c2/versions/6/. The long-term series of daily global snow depth is available at https://poles.tpdc.ac.cn/en/data/9764584a-f2df-455b-96ef-8e8968a230fa/. the long-term series of daily snow depth dataset in China is available at https://poles.tpdc.ac.cn/en/data/df40346a-0202-4ed2-bb07-b65dfcda9368/. And the 0.008° elevation data set for the TP can be found at https://data.tpdc.ac.cn/en/data/ddf4108a-d940-47ad-b25c-03666275c83a/.



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
