# Peer review of "Dynamic identification of snow phenology in the Northern Hemisphere"

_EGUsphere, 2024_

## Author Comment (AC1)

**Response to Reviewer**

Dear reviewer,

We sincerely thank you for taking the time and effort to review our paper. Your insightful feedback and constructive suggestions are invaluable and not only help to refine our research, but also deepen our understanding of the topic. We believe we have adequately addressed all of the major and minor comments, and the research has been substantially enhanced. Our point-by-point responses to your comments are listed below in black.

In this paper, the authors examine the snow phenology over the Northern Hemisphere (NH), based on satellite observations of snow cover by MODIS and IMS, and of snow depths by microwave instruments. The study focuses on snow onset, snow end date and snow cover days, as well as snow peak day.

They propose a dynamic threshold selection in constructing the snow phenology indicators, based on the local seasonal snow evolution, rather than a fixed threshold. The method has significant advantages, e.g., over the Tibetan Plateau (TP) where the snowpack can be shallow (Fig7) and the dynamic method allows for a considerably longer snow phenology (Fig9). These differences are strongly influenced by topography, and it is in the mountainous areas that the dynamic method offers the greatest benefits. In the mid and high latitudes, the difference between the two methods is small (Fig7) though, 4-5 days at most.    Over the NH overall, the differences appear small (compare Fig8 c and d), albeit it influences the indicator trends.

The study is detailed and comprehensive, and the article is fairly clearly written. It should prove a valuable study to understand the phenology and snow variability, over the TP in particular. I recommend the paper for publication provided the main comments are addressed.

Response:

We are grateful for your comments on our work, which help us to further enhance the manuscript. Our responses to your questions and comments are listed below.

**Main comments:**

1. I question the choice of treating on the same footing snow cover and snow depth (SD) in the first part of the paper. Indeed, some phenology indicators (like SCED, end of snow season) exhibit large differences exceeding one month over the TP, depending if one is considering snow cover or depth. In the 2nd part of the paper, only SD is considered. One possible way to clarify this paper is to re-structure it to address snow cover phenology (comparing the 2 instruments) in a first part, and then focus SD in a

Response:

We appreciate the reviewer's valuable and thoughtful comments.

The purpose of treating snow cover fraction (SCF) and snow depth (SD) equally in the first part is to compare whether there are significant differences in the snow phenology

45    extracted from different data and methods. Both types of snow data are commonly used for snow phenology extraction. Further, we would like to validate our idea that the extraction results of snow phenology can be significantly affected by the method. This highlights the need to improve the rationality of snow phenology extraction methods. In the second part, we select snow depth as the driving data because it shows a stable

50    single-peak pattern in each latitudinal band, reflecting the process of snow accumulation and melting. In contrast, snow cover fraction data exhibit greater randomness and heterogeneity, especially on the Tibetan Plateau (TP). Moreover, SCF data are affected by the polar night, leading to large errors north of 60°N, and cloud cover further impacts data quality. Considering these factors, we originally choose only

55    snow depth as the driving data for the second part. Here, we apply the same method to SCF data.

Since different data types describe different snow curves, the dynamic threshold for determining the snow season cannot be directly the same as the snow depth. We use the SCF data to calculate the first order derivative at each grid point and look for inflection

60    points. We find that most of the grid points change rapidly not at 10%, but more within the interval from 10% to 20%. 58.51% of the Northern Hemisphere (NH) during the snow accumulation period and 62.89% during the snow melting period fall within the 10% to 20% interval (Fig. R1). The dynamic threshold is eventually set at 15%. On the Tibetan Plateau, this threshold is significantly higher than the interval 10-20%, which

65    we hypothesize is related to the definition of the hydrological year. In this study, we define the hydrologic year as spanning from September 1 to August 31 of the following year. However, on the TP, SCF begins to grow in August. To account for this, we analyzed TP separately using a hydrologic year beginning in August and found that a

15% threshold is reasonable (Fig. R2).

[Figure]

Figure R1. Spatial distribution of thresholds extracted from the first-order derivatives using MOD10C2 data in NH. (a) Percentage thresholds associated with snow accumulation (SCOD) from the first half of the first-order derivative maximum value. (b) Percentage thresholds associated with snow melting (SCED) from the last point of half of the first-order derivative minimum value.

[Figure]

Figure R2. Intra-annual variability in the normalized snow cover fraction in the Tibetan Plateau. The gray curves represent the original SCF, the red curves represent

the 30-day smoothed SD, and the yellow curves represent the first-order derivative.

The unit DOHY is an abbreviation for day of the hydrological year, defined as August 1 through July 31 of the following year.

We normalize and plot the intra-annual SCF change curves for the four latitudinal bands on the same graph for comparison and find that the snow curves become wider with increasing latitude, which implies that the snow season is lengthening. In contrast to the other latitudinal bands, the snow curve of TP fluctuates a lot. The 15% position is a good match to the inflection point of the SCF curve. Employing the 15% dynamics threshold method, we extract SCF thresholds in the NH. Except for TP, the new thresholds for the entire NH are lower than the traditional fixed thresholds, implying that the traditional thresholds may result in an underestimation of the snow season.

[Figure]

**Figure R3.** The dynamic threshold of the MOD10C2 dataset across the NH. (a) Intra-annual variability in the normalized MOD10C2 data for four latitudinal zones. Shading represents the interval 10%-20%, and the dashed line represents the dynamic threshold of 15%. The actual maximum snow cover fraction for each latitude band is in parentheses. The unit DOHY is an abbreviation for day of the hydrological year. (b)

Spatial distribution of multi-year average snow cover fraction thresholds in the NH extracted using the snow dynamics threshold method.

100

The spatial distributions of the snow phenology indicators extracted using the snow dynamics threshold method are similar to those of the original method with increasing latitude and elevation, snow cover duration (SCD) lengthens, snow cover onset day (SCOD) advances, and snow cover end day (SCED) delays. The use of the new

105 method has resulted in a generally longer snow season due to the lowering of the thresholds. SCED demonstrated greater differences compared to SCOD.

[Figure]

**Figure R4.** Spatial distribution of snow phenology extracted by MOD10C2 data over

110 the NH for the hydrological years 2000-2018. (a) Multiyear averaged snow cover duration (SCD), (b) snow cover onset day (SCOD), (c) snow cover end day (SCED) extracted by the fixed threshold method. (d, e, f) Same as (a, b, c) but for the dynamic threshold method.

115 In order to show the difference between the snow phenology extracted by the dynamic threshold method and the fixed threshold method more clearly, we perform statistics in four latitudinal bands (including TP) and in the whole NH. We find that similar to the results for snow depth, the largest difference in snow phenology remains in the TP, where the SCD differs by 86 days. And the main difference in SCD is caused by SCED,

120 with a smaller contribution from SCOD.

[Figure]

**Figure R5.** Comparative radar maps of (a) snow cover duration (SCD), (b) snow cover onset day (SCOD), (c) snow cover end day (SCED) extracted by the snow dynamic threshold method and the traditional fixed threshold method in five regions. The solid red line represents the dynamic threshold method, and the blue dashed line represents the fixed threshold method.

In summary, our results using SCF data for snow phenology method improvements show agreement with those based on SD. The dynamic thresholds for both datasets fall within the 10%-20% range, and both methods extend the snow season at low latitudes. The results are similar for both data, and there are flaws in the SCF data, such as the effects of polar night.

We have included the SCF results in Section 3.2 and Discussion section, and placed the corresponding graphs in the supplement. This preserves the original structure of our study—first comparing differences in existing snow phenology extraction methods and then improving them. Moreover, the results of SCF can be shown to prove the universality of the methods.

Line 348:

*Given the limitations of MOD10C2 data, such as its susceptibility to polar night effects and fluctuations, we select SD as the primary driving data for this study. However, to demonstrate the robustness of the dynamic threshold method, we also apply MOD10C2 data for dynamic snow phenology extraction (see supplement). Our results indicate that for MOD10C2 data, a dynamic threshold of 15% is more appropriate. After applying the dynamic threshold method, the snow phenology results closely align with those*

*obtained from SD, exhibiting longer SCD, earlier SCOD, and later SCED in mid- and low-latitude regions. The most pronounced discrepancies are observed over the TP. However, since snow cover data are influenced by the polar night at high latitudes, direct comparisons cannot be made at these latitudes.*

150

Line 449:

*Since the accuracy of passive microwave detection increases with snow depth, the passive microwave remote sensing data is more effective for analyzing snow phenology in regions with consistent snow cover (Armstrong & Brodzik, 2001; Savoie et al.,2009).*

155 *In areas with shallow snow with wet snow, the accuracy of passive microwave remote sensing data is reduced, and the snow depth indicator may not accurately capture accumulation and melting processes. In addition, for the transient snow area, the snow depth curve is more volatile, which makes the assumed single-peak structure untenable. After comprehensive consideration, the snow cover fraction may be a more reliable*

160 *indicator in such cases. Therefore, we perform another extraction of dynamic snow phenology using the snow cover fraction data, and the results are similar to SD, but with greater differences in TP (see supplement).*

2. I am concerned about the methodology at locations where the snow curve is not

165 monotonous across the season and has several maxima and minima linked to episodic snowfall and melt. Smoothing or climatological averaging should alleviate this potential problem. This seems be the case for the snow cover data over the TP (Fig4). This restricts the applicability of the method, and the authors expressed this concern (L445-446), even for SD at some locations over the TP where the snow layer is

170 shallow.  Please discuss this issue in the Methodology section.

Response:

Thank you for your suggestion. As you mentioned, the occurrence of sporadic snowfall events results in snow curves that are not strictly unimodal throughout the seasonal cycle, particularly in regions with unstable snow conditions. This underscores the

importance of applying smoothing techniques in our approach to mitigate the influence of noise. To determine the smooth window, we analyze the dynamic threshold using percentage of snow depth extracted from the first-order derivatives under different smoothing windows. Snow depth percentage stabilizes when the smoothing window reaches around 30 days. The curve drops sharply for smaller windows but fluctuates minimally beyond this point. And snow on the lunar scale is more stable and reliable, less affected by disturbances. Thus, a 30-day smoothing window is chosen. Following your suggestions, we have extended the discussion on this issue in the Methodology section.

[Figure]

**Figure R6**. The relationship between the percentage of snow depth (dynamic threshold) and smooth window in (a) the Northern Hemisphere, (b) the Tibetan Plateau, (c) 30°N–40°N, (d) 40°N–50°N, (e) 50°N–60°N, and (f) 60°N–75°N. The black line is the original line, the black dot is the specific value for each year, and the red line is the trend line.

Line 143:

*Occasional snowfall events, such as short-duration or localized snowfall, can introduce anomalous fluctuations in the snow curve, leading to multiple small peaks or atypical maxima. These fluctuations represent short-term meteorological noise rather than the long-term seasonal evolution of the snow cover. This is particularly common, especially*

*in unstable snow areas (e.g., the Tibetan Plateau). Smoothing can reduce these instabilities and make the snow curve more reflective of seasonal snowpack changes.*

**Minor comments:**

1. L 36: Concerning the decreasing length of the snow season in Notarcola (2022): wasn't this paper focused only on the mountainous regions?

Response:

Thank you for the insightful comment. The study by Notarnicola primarily focuses on mountainous regions, and its conclusions regarding the shortening of the snow season are also specific to these areas. This was not explicitly clear in our original text. We have now revised the manuscript to clarify this point.

Line 35:

*Moreover, the length of the snow season and the number of snow days are shortened in global mountain regions (Notarnicola et al., 2022).*

2. L98: It is a bit unclear what the authors mean by "replacing the dataset"?

Response:

The global snow depth data set of Che et al. is affected by satellite orbits, leading to substantial missing measurements at low and middle latitudes, especially on the Tibetan Plateau. The location of the missing data varies from day to day with no regularity. In the other data set of Che et al. on snow depth in China, an algorithm is used to fill in the missing data. Given that the mid- and low-latitude regions are representative of unstable snow conditions, they are the primary focus for improving snow phenology using our dynamic threshold approach. However, the presence of significant data gaps introduces uncertainties in snow phenology extraction that need to be addressed. To mitigate this, we replace the snow depth data for China in the global dataset with Chinese snow depth data, ensuring more accurate phenology representation in this region.

In order to give the reader a better understanding, we have changed the sentence in the

manuscript.

*The long-term series of daily global SD is affected by satellite orbits, leading to substantial missing measurements at low and middle latitudes. To minimize the negative effects of data gaps, we substitute the China region in the global dataset with another set of snow depth data for the China region (Che et al., 2015).*

3.L106: What is SCE and where is it defined ? Or is it a Typo and should it be SCA as it refers to IMS ?

Response:

Thanks for your suggestion. This should indeed be changed to SCA as the IMS data can represent changes in SCA, which have been changed in the manuscript.

*For the daily IMS binary SCE dataset, no additional processing is required to determine if the grid is covered with snow.*

4.IMS also measures snow cover fraction: hence the labels (e.g. in Table 2, figures 3-4) should be more consistent: using IMS, SCF, SD mixes instruments names and the variables.

Response:

Thank you for your valuable suggestion. As you pointed out, IMS and MOD10C2 represent different forms of SCF, and using SCF to refer to MOD10C2 may cause confusion. To clarify this, we have made the following adjustments: the IMS label remains unchanged in all figures, while SCF has been replaced with MOD10C2 to explicitly refer to the dataset. This revision ensures a clearer distinction between the two data sources.

5.L145: The first derivative also goes to zero at the maximum of the snow curve.

Response:

Thank you for your reminder. Indeed, the first derivative corresponding to the maximum of the snow curve is also zero. To avoid confusion, the following annotations have been added to the manuscript.

Line 151:

*When the first-order derivative equals zero (at the beginning and end of the curve, not the maximum), it shows that snow has not started accumulating or has completely melted.*

6.When adapting the formula from the vegetation index, is it true that SnowMin is actually zero throughout this study?

Response:

$Snow_{min}$ is not universally zero; rather, it is determined by local snow conditions. In high-altitude regions with perennial snow cover, $Snow_{min}$ remains nonzero, whereas in mid- and low-latitude regions with relatively sparse snowfall, most $Snow_{min}$ values approach zero. This spatial variability is analogous to vegetation indices, which may not be zero depending on the specific local environment.

7.A couple of points are not clear in the Methodology: on one hand, "The above process is carried out for each grid in the NH (hence locally), yet above, it seems that the ratio is defined in "latitudinal zones" (implying a zonal average). The authors also mention multi-annual averages of the threshold, which implies that the threshold is defined for each year and then averaged, as opposed to using a climatological evolution to define a threshold. Please clarify.

Response:

For the first question, to assess the generalizability of the ratio, we conduct calculations not only for the entire NH and for individual latitude bands but also at each grid point (Figure 5). The results for the entire NH are presented in Figure 1, while those for the different latitudinal analyses are shown in Figure R6. The ratios are found to converge towards approximately 10%. When performing the same calculation at each grid point,

we find that 73.05% and 82.65% of the two ratios fell within the 5%-15% range,
respectively. Consequently, the final ratio is set at 10%, ensuring its validity both at the
hemispheric scale and latitudinal zones, as well as at the local level.

For the second question, as you mentioned, the thresholds are defined annually, and the
snow phenology for each year is extracted based on that year's specific threshold. This
approach allows both the interannual variations in the thresholds and snow phenology
to reflect climate evolution. For example, as the climate warms, changes occur in the
maximum snow accumulation, the timing of snowfall, and snow duration, leading to
shifts in the snow curve pattern. Consequently, the thresholds derived from the snow
curves also vary, resulting in corresponding interannual changes in extracted snow
phenology. In the manuscript, the thresholds are averaged over multiple years solely for
spatial comparison with the traditional 2 cm threshold. The results indicate substantial
variations in thresholds across different latitudes, suggesting that a uniform threshold
is not appropriate. However, in the subsequent process of snow phenology extraction,
the multi-year averaged threshold was not used. Instead, year specific thresholds are
applied to ensure that the influence of climate evolution is not masked.

[Figure]

**Figure R7**. Schematic diagram of ratio results calculated at different latitudinal zones,
including the Tibetan Plateau, 30°N–40°N, 40°N–50°N, 50°N–60°N, and 60°N–75°N.
Similar to Figure 1.

8.Caption of Fig 4 over which years is this intra-annual (climatological?) variation established?

Response:

This represents the intra-annual variation curve averaged over the period from 2000 to 2018. The original figure caption do not specify this information, now it has been added.

Line 222:

*Intra-annual variations of IMS, MOD10C2, and SD in five latitudinal zones (including the Tibetan Plateau) from 2000 to 2018.*

9.L242-265: there are lot of repeats with the Methodology section 2.3

Response:

Thank you for your advice. We have condensed this section to try to avoid repeating sentences.

Line 140:

*To identify the optimal $Snow_{radio}$, we normalize and smooth the interannual SD variation curves for the entire NH and each latitudinal zone (including the Tibetan Plateau) with a 30-day moving window and then calculate their first-order derivatives. Occasional snowfall events, such as short-duration or localized snowfall, can introduce anomalous fluctuations in the snow curve, leading to multiple small peaks or atypical maxima. These fluctuations represent short-term meteorological noise rather than the long-term seasonal evolution of the snow cover. This is particularly common, especially in unstable snow areas. Smoothing can reduce these instabilities and make the snow curve more reflective of seasonal snowpack changes. We assume the curve has a single peak structure. The first-order derivative shows the rate of snow accumulation or melting, with its extreme points indicating the steepest changes. However, when the first-order derivative equals zero (beginning and end of the curve, not the maximum), it shows that snow has not started accumulating or has completely melted. The intermediate state between the maximum rate and no change represents when snow starts to accumulate or melting is nearly complete, which is what we are looking for in*

*SCOD and SCED. So, here we simply choose the extreme midpoint of the first-order derivative as the snow curve turning point. The percentage of tthe urning point is the threshold we need. The above process is carried out for each grid in the NH, and Figure 1 shows a schematic of the entire NH. The percentages for SCOD and SCED fall between 5% and 15% (marked by red circles). This threshold range (i.e. $Snow_{ratio}$) can therefore serve as an indicator for the snow season.*

10.The early onset of snow annual cycle for the TP is interesting; is it governed by the high-altitude areas? It might be worthed to split this Fig 6 into mountainous and non-mountainous areas, like done in the later part of the paper.

Response:

This is an interesting and complex issue. The early onset of the snow season on the Tibetan Plateau (TP) is a complex phenomenon influenced by multiple factors. The high-altitude environment of TP, with an average elevation exceeding 4,000 m, plays a crucial role in this process. At such elevations, lower temperatures prevail throughout the year, facilitating an earlier transition into winter and creating favorable conditions for snowfall. Additionally, the region's complex topography contributes to the retention of cold air, further enhancing local cooling effects (Yang et al., 2014). Moreover, TP is subject to the combined influence of the East Asian winter monsoon and the South Asian monsoon. During autumn, the intensification of the East Asian winter monsoon enhances cold air advection, leading to frequent southward intrusions of cold air masses. This process results in a rapid decline in temperature, promoting an earlier onset of snowfall (Wu et al., 2012). In short, the early snow season in TP is the result of multiple factors, and its high altitude makes a considerable contribution.

Taking your advice, I have divided Figure 6 into mountain ranges and non-mountain ranges and added it to Figure 6 in the manuscript with the appropriate captions. Regardless of the latitudinal belt, the snow curve in the non-mountainous region would be narrower than that in the mountainous region, implying a shorter snow season in the non-mountainous region. An interesting phenomenon is that the peak of non-mountain

range snow arrives sooner. An interesting phenomenon is that the peak of non-mountain range snow arrives earlier in TP. Snow on TP mountains exhibits greater stability and shares characteristics with high-latitude snow, leading to a snow curve that closely resembles that of high-latitude mountainous regions. However, the snow season in the TP non-mountains is significantly earlier than in other latitudinal belts. This may be related to the fact that TP has more occasional random snowfalls, and the snow is stored due to colder temperatures. This is a very interesting phenomenon, and we will follow up by paying more attention to this phenomenon and conducting a more detailed study.

[Figure]

**Figure R8.** The dynamic SD threshold across the Northern Hemisphere. Intra-annual variability in the normalized SD for five latitudinal zones in (a) the whole Northern Hemisphere, (c) the Northern Hemisphere mountain ranges, (d) the Northern Hemisphere non-mountain ranges. Shading represents the interval 5%-15%, and the dashed line represents the dynamic threshold of 10%. Actual maximum snow depths for each latitude band are in parentheses. The unit DOHY is an abbreviation for day of the hydrological year, defined as September 1 through August 31 of the following year. (b) Spatial distribution of multi-year average snow depth thresholds in the Northern Hemisphere extracted using the snow dynamics threshold method.

Line 284:

*Defining SDtopo greater than 200 as a mountain range divides the Northern Hemisphere into mountain ranges and non-mountain ranges. Regardless of the*

*latitudinal belt, the snow curve in the non-mountainous region would be narrower than that in the mountainous region, implying a shorter snow season in the non-mountainous region. Snow curves in mountainous regions are more stable and show the same pattern in the five latitudinal zones. Therefore, the location of the 10% threshold is appropriate in both mountain range and non-mountain range areas where the turnover change occurs.*

References:

Wu, G., Liu, Y., He, B., Bao, Q., Duan, A., & Jin, F.-F: Thermal Controls on the Asian Summer Monsoon. Scientific Reports, 2(1), 404. https://doi.org/10.1038/srep00404, 2012.

Yang, K., Wu, H., Qin, J., Lin, C., Tang, W., & Chen, Y: Recent climate changes over the Tibetan Plateau and their impacts on energy and water cycle: A review. Global and Planetary Change, 112, 79–91. https://doi.org/10.1016/j.gloplacha.2013.12.001, 2014.

**Typos & English.**

1. L125 (and many other places) ratio not radio

Response:

Thank you for your correction, all statement errors in the manuscript have been corrected.

Line 125:

*When the $NDVI_{ratio}$ exceeds is below a certain threshold, the corresponding day of the year is determined as the EOS.*

2. Caption of Fig 1 : "has not started … has ended"

Response:

The sentence has been modified.

Line 137:

*Gray shading indicates the snow season has not started or has ended, blue shading indicates the snow accumulation period, and red shading indicates the snow melting*

*period.*

3.  L217: snow conditions at given grid points

Response:

420    We have revised the sentence.

Line 217:

*Specifically, when the threshold is reduced, snow conditions at given grid points are more easily reached, resulting in a longer SCD, earlier SCOD, and later SCED.*

425    4.Caption of Fig 4 : replace …"snow elements" by the annual snow maxima over the respective areas.

Response:

The sentence has been modified.

Line 224:

430    *The values in the graphs characterize the annual snow maxima over the respective areas.*

---

## Author Comment (AC2)

**Response to Reviewer**

Dear reviewer,

5    We are very grateful for your recognition and encouragement of our work, and your professional guidance has provided us with new perspectives and ideas that will undoubtedly enhance the quality and impact of our research. We believe we have addressed all of the comments, and we feel our manuscript has become substantially stronger as a result of these improvements. Our point-by-point responses to your

10   comments are listed below in black.

The manuscript presents a valuable contribution to the field of snow phenology by proposing a dynamic threshold method for snow phenology extraction in the Northern Hemisphere. The study effectively highlights the limitations of the traditional fixed threshold approach and demonstrates the advantages of the dynamic method in capturing the spatial heterogeneity and temporal variability of snow cover.

Response:

Thank you very much for your insightful and thorough evaluation of our manuscript. We greatly appreciate your recognition of our work.

**Main comments:**

1.The sentence "Our analysis further indicates that the changes in snow depth exhibits a significant shift around 10% of peak value across the Northern Hemisphere, marking the transition between the snow and non-snow seasons." need to clarity. I feel it's difficult to understand.

Response:

We apologize for our lack of clarity, which makes it difficult for readers to understand. This has now been corrected.

Our main approach is presented in Section 2.3 and illustrated in Figure 1. Specifically, after normalizing the snow depth data, we compute its first derivative and determine the percentage threshold based on the physical meaning of the first-order derivative. The first-order derivative represents the actual rate of snow accumulation and melting, with its extreme points indicating the maximum rate of snow change. When the first-order derivative equals zero (at the beginning and end of the curve, rather than at the maximum), it signifies that snow has either not yet begun to accumulate or has completely melted. The intermediate state between the maximum rate of snow change and no change corresponds to the onset of snow accumulation or the near completion of melting—this is the key phase we seek to identify for SCOD and SCED. Therefore, we define the extreme midpoint of the first-order derivative as the turning point of the snow curve. The percentage value at this turning point serves as the threshold required

40 for our analysis.

To evaluate the generalizability of the percentage threshold, we perform calculations not only for the entire Northern Hemisphere and individual latitude bands but also at the grid-point level. The results for the Northern Hemisphere are presented in Figure 1, while those for different latitudinal bands are shown in Figure R1. The ratios

45 consistently converge toward approximately 10%. When applying the same calculation at each grid point, we find that 73.05% and 82.65% of the two ratios fall within the 5%-15% range, respectively (Figure 5). Therefore, the final ratio is set at 10%. It is clear from the schematic in Figure 1 that below the 10% position of the curve, the curve is very flat. Above the 10% position, the curve changes rapidly.

50 Due to the word limit of the abstract, it cannot be interpreted in this way with multiple characters. We have changed this sentence in the manuscript for better understanding.

Line 11:

*After normalizing, the percentage snow depth curve turns significantly at the 10% position, marking the transition between the snow and non-snow seasons.*

55

[Figure]

[Figure]

**Figure R1**. Schematic diagram of ratio results calculated at different latitudinal zones, including the Tibetan Plateau, 30°N–40°N, 40°N–50°N, 50°N–60°N, and 60°N–75°N. Similar to Figure 1.

60

2. The introduction could be more focused on the specific research gap addressed by the study. Could you rewrite it?

Response:

Thank you for your advice. Indeed, the introductory part of our manuscript has too much padding, and the specific research gap is told relatively little later. Therefore, we have abbreviated the background and detailed the problems in the field of snow phenology in the penultimate paragraph. The section of the manuscript on the specific research gap, which has been significantly altered, is shown below.

Line 53:

*Snow phenology was generally obtained through a two-step process in previous studies, i.e., identifying the presence or absence of snow in the grid based on a given threshold and calculating snow phenology indicators (Peng et al., 2013; Yang et al., 2019; Notarnicola, 2020). Various types of snow data are used to extract snow phenology, including SDs, snow cover fractions, and snow water equivalents, leading to possible differences in identified snow phenology (Chen et al., 2015; Guo et al., 2022). Additionally, most studies have employed a fixed threshold to extract snow phenology in different regions and years (Brown et al., 2007; Gao et al., 2011; Yue et al., 2022; Tang et al., 2022). The fixed threshold for snow phenology fails to account for the variations in snow cover across the NH. In fact, snow cover increases with latitude, with thick and stable snow cover at high latitudes and shallow and short-lived snow cover at middle and low latitudes, especially in the TP (Orsolini et al., 2019). In addition, the snow changes from year to year due to many aspects of the climate, and the regional snow cover trends exhibit a heterogeneous and non-linear response to its regional warming rate (Blau et al., 2024). Snow conditions are variable, but thresholds are always fixed, which can lead to inaccuracies of snow phenology. Therefore, employing different snow data and a fixed threshold will lead to uncertainties in extracted snow phenology. At present, it has been proven in the methods of extracting vegetation phenology that fixed thresholds cannot accommodate spatio-temporal heterogeneity, ignore inter-annual variations, and are not applicable to diverse*

*vegetation types, among a series of other problems (White et al., 1997; Mo et al., 1997).*
*However, this issue regarding snow phenology extraction methods has not yet received*
*attention and resolution. In fact, the onset of the snow season is marked by the sustained*
*accumulation of snowfall as ground snow, rather than being determined by a fixed*
95 *threshold. We aim to propose a novel method that incorporates both spatial*
*heterogeneity in snow cover and temporal variability to extract snow phenology,*
*reducing the uncertainty associated with the fixed threshold method from a physical*
*meaning perspective.*

100 3.If possible, could you add a sentence or two to explicitly state the research objectives
and the main contributions of the study.

Response:

Thanks for your suggestion. We have added sentences to explicitly state the research
objectives and the main contributions of the study.

105 Initially, we find that the method commonly used to extract snow phenology overlooks
regional variability in snow characteristics and its temporal evolution. This limitation
motivates us to improve the approach to enhance its accuracy and applicability. To
support our argument, we first compare snow phenology results derived from different
datasets to assess whether the fixed threshold used in snow evolution can reasonably
110 capture the start and end of the snow season. After highlighting the shortcomings of the
existing method, we aim to refine the method to achieve more accurate snow phenology
extraction. Finally, we focus on the Tibetan Plateau, a region with unique topography
and climate, to examine the generalizability and limitations of the new method.

In the manuscript, it is true that our objectives are not explicitly written, and they have
115 now been added.

Line 68:

*We aim to propose a novel method that incorporates both spatial heterogeneity in snow*
*cover and temporal variability to extract snow phenology, thereby reducing the*
*uncertainty associated with the fixed threshold method.*

The main contribution of our study is the completion of our predefined objectives. At the end of the Conclusions and Discussion in the manuscript, we briefly summarize our main contributions of the study.

Line 413:

*In this study, we explore the spatial distribution of snow phenology in the Northern Hemisphere (NH) using several sets of satellite remote sensing snow data and multiple methods. A new extraction method for snow phenology is proposed in the NH, and the differences in snow phenology using the traditional and new methods are compared to evaluate the new snow phenology method.*

4.It would be helpful to provide a brief explanation of the rationale behind choosing the 30-day moving window for smoothing the snow depth data.

Response:

Thank you for your advice. In practice, the selection of a 30-day moving window is not of particular significance; its main function is to smooth the data, reducing noise and limiting the influence of random snowfall timing on snow phenology extraction. Snow changes are more stable and reliable on a monthly scale. To evaluate the impact of different smoothing windows, we calculate the dynamic threshold using percentage of snow depth derived from the first-order derivatives (see Methods) and illustrate the results in a line graph (Fig. R2). The snow depth percentage remains relatively stable when the smoothing window approaches 30 days. When the window is smaller than 30 days, the curve declines steeply, whereas for values exceeding 30 days, fluctuations are minimal. Based on this analysis, we adopt a smoothing window of 30 days. While slight variations in the window size do not significantly alter the results, a window that is too small amplifies noise, whereas an excessively large one may obscure valid information.

[Figure]

**Figure R2**. The relationship between the percentage of snow depth (dynamic threshold) and smooth window in (a) the Northern Hemisphere, (b) the Tibetan Plateau, (c) 30°N–40°N, (d) 40°N–50°N, (e) 50°N–60°N, and (f) 60°N–75°N. The black line is the original line, the black dot is the specific value for each year, and the red line is the trend line.

5.The conclusions clearly summarize the main findings and contributions of the study.

Response:

Thank you for your compliments. We have followed your suggestions and further improved the manuscript.

---

## Author Comment (AC3)

**Response to Reviewer**

Dear Dr. Xiao,

5    We sincerely appreciate your time and effort in discussing our study and sharing your insightful suggestions. Your feedback has been incredibly valuable in deepening my understanding of the topic and identifying areas for improvement. Our point-by-point responses to your comments are listed below in black.

**Main comments:**

1.According to numerous previous works and my analysis (https://doi.org/10.1016/j.isprsjprs.2024.07.018), the uncertainty in snow phenology is mainly derived from the accuracy of daily snow cover products. You always mentioned there are potential bias when using the fixed threshold to determine snow phenology. Through your whole text, you didn't specifically illustrate this type of uncertainties. May I ask you to give some examples to clarify this uncertainty what you mentioned?

Response:

We agree with your perspective that uncertainty in data will pass to snow phenology calculation. In this study, we did not explicitly quantify the bias because there are no direct observational data for snow phenology. Facing snow data uncertainty and a lack of snow phenology references, we try to optimize the methods for snow phenology extraction.

We do not mean that the traditional fixed threshold approach is incorrect, but we identify the limitations of the traditional fixed threshold method and explore possible improvements. Spatially, snow conditions vary across regions. Applying a uniform threshold implies treating all snow conditions as identical, which is unreasonable. Snow conditions also differ significantly over time, such as from the Industrial Revolution to the present.

We think that when snowfall can be steadily converted into snow on the ground, it means the onset of the snow season. In this work, we define the inflection point of snow coverage or depth curve, serving as a dynamic threshold for identifying the start of the snow season. In contrast, a fixed 2 cm or 50% threshold will appear as different phases of the snow season in different regions and time periods. Similarly, snow cover extent data are obtained via pre-processing, such as the sub-grid snow information extraction, which may introduce several thresholds or criteria.

As mentioned above, we have recognized the impact of data uncertainty on snow phenology identification, but we believe that the spatial and temporal heterogeneity of snow cover needs to be considered. This work is a preliminary exploration of dynamic

snow phenology, and we hope to further refine it in future work, including your important suggestions. According to your comment, we revise our manuscript to further clarify the above statement.

---In the Abstract, "Previous studies commonly employed … leading to potential biases of snow phenology." This description is too general.

Response:

Thank you for the suggestion. In this sentence, we want to express the limitation of fixed thresholds. We have revised this sentence for the better understanding of the readers.

Line 7:

*Previous studies commonly employed fixed threshold methods to extract snow phenology, which cannot represent the differences in the beginning/end of the snow period under different snow conditions in the Northern Hemisphere, leading to potential uncertainties of snow phenology.*

--- Lines 65-66, "The fixed threshold for snow phenology fails to account for the variations in snow cover across the NH". Please give us more explanation on this. Do you have any related references to support your statement? Additionally, the following sentence "In fact, snow cover increase …. Especially on the TP" cannot support your statement on the uncertainties due to using fixed threshold.

Response:

As in the previous responses, we believe that a fixed threshold fails to represent the actual beginning/end of snow cover under different snow conditions, particularly when considering long-term climatic shifts and regional differences in snow.

We propose that the initiation of the snow season should be determined by identifying the point at which snowfall consistently leads to stable accumulation on the ground. In our approach, we define this using a dynamic threshold based on the inflection point of the snow coverage or depth curve. This inspiration comes from the methodology used

in the extraction of vegetation phenology. It is well known that snow and vegetation in the Northern Hemisphere exhibit similar large-scale spatial patterns, both varying with latitude. However, while vegetation coverage decreases with increasing latitude, snow cover follows the opposite trend, expanding at higher latitudes. Additionally, both snow and vegetation are influenced by climate change, leading to temporal variations in their distribution and dynamics. The word "phenology" was originally derived from vegetation and was subsequently extended to include snow phenology. Consequently, methods for extracting vegetation phenology are more advanced and well-developed compared to those for snow phenology. At present, it has been proven in the methods of extracting vegetation phenology that fixed thresholds cannot accommodate spatio-temporal heterogeneity, ignore inter-annual variations, and are not applicable to diverse vegetation types, among a series of other problems (White et al., 1997; Mo et al., 2012). However, this issue regarding snow phenology extraction methods has not yet received attention and resolution. Building on insights from vegetation studies, we hypothesize that similar principles apply to snow. The fixed threshold method, however, fails to account for the temporal and spatial variability of snow, limiting its effectiveness in capturing dynamic snow processes.

We have added to the manuscript in order to make it more comprehensible to the reader.

Line 65:

*The fixed threshold for snow phenology fails to account for the variations in snow cover across the NH. In fact, snow cover increases with latitude, with thick and stable snow cover at high latitudes and shallow and short-lived snow cover at middle and low latitudes, especially in the TP (Orsolini et al., 2019). In addition, the snow changes from year to year due to many aspects of the climate, and the regional snow cover trends exhibit a heterogeneous and non-linear response to its regional warming rate (Blau et al., 2024). Snow conditions are variable, but thresholds are always fixed, which can lead to inaccuracies of snow phenology. Therefore, employing different snow data and a fixed threshold will lead to uncertainties in extracted snow phenology. At present, it has been proven in the methods of extracting vegetation phenology that fixed thresholds*

*cannot accommodate spatio-temporal heterogeneity, ignore inter-annual variations, and are not applicable to diverse vegetation types, among a series of other problems (White et al., 1997; Mo et al., 2012). However, this issue regarding snow phenology extraction methods has not yet received attention and resolution. In fact, the onset of the snow season is marked by the sustained accumulation of snowfall as ground snow, rather than being determined by a fixed threshold. We aim to propose a novel method that incorporates both spatial heterogeneity in snow cover and temporal variability to extract snow phenology, reducing the uncertainty associated with the fixed threshold method from a physical meaning perspective.*

References:

Mo, J., Zhu, W., Wang, L., Xu, Y., and Liu J.: Evaluation of remote sensing extraction methods for vegetation phenology based on flux tower net ecosystem carbon exchange data. Chinese Journal of Applied Ecology, 23(2), 319-327. https://doi.org/10.13287/j.1001-9332.2012.0072, 2012.

White, M. A., Thornton, P. E., and Running, S. W.: A continental phenology model for monitoring vegetation responses to interannual climatic variability. Global Biogeochemical Cycles, 11, 217–234. https://doi.org/10.1029/97GB00330, 1997.

--- Lines 68-69: "Snow conditions are variable, … underestimation or overestimation of snow phenology." Please add more explanations on this underestimation and overestimation. Is there any published works to support this statement?

Response:

It is a misrepresentation here. As mentioned in your first question, there are no reference data to support our assessment of whether snow phenology is underestimated and overestimated.

Therefore, we have revised the description to highlight our consideration of the impact of spatial and temporal heterogeneity of snow in the Northern Hemisphere on snow phenology. Please see the answer to the previous question for more details.

2.Line 14-15: "At low and middle latitudes, the snow cover duration (SCD) extends, the snow cover onset day (SCOD) advances, and the snow cover end day (SCED) delays, …" This conclusion is quite different from what we got up to now. Does your analysis conclusion tell us we have more snow in this region in a warming world?

Response:

We apologize for any ambiguity in my previous explanation, which may have led to a misunderstanding. The comparison of extending and advancing here specifically refers to the traditional fixed threshold method. In particular, compared to the conventional 2 cm threshold, the dynamic threshold method results in a longer snow cover duration (SCD), an earlier snow cover onset date (SCOD), and a later snow cover end date (SCED) at low and mid-latitudes, especially over the Tibetan Plateau, where the SCD difference can reach up to 28 days. Conversely, at higher latitudes, the changes follow the opposite pattern. Therefore, the reference here does not pertain to absolute changes in time scales but rather to the relative differences in snow phenology results derived from the two methods. To prevent misinterpretation, clarifications and additional labels have been incorporated into the manuscript.

Line 13:

*Using the dynamic threshold method, there is an earlier snow cover onset day (SCOD), a later snow cover end day (SCED), and a longer snow cover duration (SCD) at low and middle latitudes, especially on the Tibetan Plateau, where the SCD differences can reach 28 days. The differences in snow phenology at higher latitudes are reversed.*

3.MOD10C2 products provide a maximum snow cover extent during this 8-days period. That means you would potentially overestimate the snow cover phenology metrics (snow cover days, snow onset date and snow end date) when you used this data. "For the SCF dataset, … after the last identified snow cover." (Line 112-113) Please cite references to support this processing's reasonability.

Response:

Thank you for your question. As you mentioned, a major limitation of optical remote

sensing is cloud contamination, which obscures snow data on most days during the snow season. To mitigate this issue, 8-day and 16-day composite snow cover products are generated to reduce cloud interference (Hall et al., 2002). The MOD10C2 data is one of the products of optical remote sensing, which provides the maximum snow cover extent over an 8-day period. Consequently, using it for snow phenology extraction may cause an overestimation of the snow season. In order to compare snow phenology results from different data, we chose this data set with reference to Chen et al. (2015). However, the bias that this data set brings to snow phenology is not negligible. Therefore, we will follow up by choosing daily-scale, more accurate SCF data for the extraction of phenology. The method used in L112 is adopted from the study by Chen et al. (2015). The citations have now been added to the manuscript.

Line 112:

*For the MOD10C2 dataset, considering the 8-day temporal resolution, the SCOD is the date four days before the first identified snow cover, and the SCED is four days after the last identified snow cover. SCD is determined by multiplying the number of snow occurrences by eight (Notarnicola, 2020; Yue et al., 2022; Guo et al., 2022; Chen et al., 2015).*

References:

Chen, X., Liang, S., Cao, Y., He, T., and Wang, D.: Observed contrast changes in snow cover phenology in northern middle and high latitudes from 2001–2014. Scientific Reports, 5. https://doi.org/10.1038/srep16820, 2015.

4.1: which part of data do you use to plot this figure? 1989-2018 or 2000-2018? Single data and which data? Additionally, this curve is like for vegetation growth instead of snow cover evolution. Please check it again. The fact is that Summer is no snow (DOY=180 days). Finally, how can I understand the term "percentage of snow depth"? For your reference, here is a in-situ observation of snow evolution within a snow cover year (10/1 – 9/31) https://www.climatehubs.usda.gov/hubs/northwest/topic/30-year-normals. Regarding only snow depth variable used in your method, I am confused how

you use snow cover products (IMS and SCF) to calculate snow phenology metrics.

185  Response:

Thank you for your question. We will address each point individually.

1. The snow depth data used to generate this figure covers the period from 2000 to 2018. I apologize for any ambiguity in my previous explanation that may have led to a misunderstanding. In fact, the figure presents results based on the hydrological year, which spans from September 1 to August 31 of the following year. As a result, Day 180 falls within the winter season, leading to a high SD value. To avoid such confusion, I have revised "DOY" in the text to "DOHY" (day of the hydrological year), which I hope will improve clarity.

2. The label "percentage of snow depth" represents the value after normalizing for snow depth. The schematic equation is as follows.

$$percentage\ of\ snow\ depth = \frac{Snow\ depth - Snow\ depth_{min}}{Snow\ depth_{max} - Snow\ depth_{min}}\ , \tag{1}$$

3. In Section 3.1, Snow depth is selected as the driving data because it consistently follows a stable single-peak pattern across different latitudinal zones, effectively representing the accumulation and melting processes of snow. In contrast, snow cover data are more irregular and spatially diverse, especially over the Tibetan Plateau (TP). Furthermore, snow cover measurements are significantly impacted by the polar night, causing considerable inaccuracies north of 60°N, while cloud interference further compromises data reliability. Therefore, we use the snow depth dataset to improve the snow phenology extraction method. In fact, the dynamic threshold method can also be applied to SCF data. The results of the new method using a dynamic threshold for SCF are nearly similar to those of SD, and we show the results for SCF in the Appendix. However, IMS data have already undergone preprocessing. Since their thresholds cannot be adjusted, it is not possible to apply a dynamic threshold to IMS data.

210  5.Through your manuscript, you used a 5% or 10% to determine snow phenology metrics. Is this another type of "fixed threshold" method. Additionally, you didn't include some comparisons, validations, and evaluations. How do you convince our

215 Response:

We agree with your point of view. We acknowledge that the dynamic threshold method for snow phenology is not fully dynamic; however, it is relatively more dynamic compared to the traditional fixed threshold method. Specifically, the fixed threshold approach fails to account for interannual variations and spatial differences in snow,

220 treating snow uniformly across all locations and time periods without distinction, which is evidently unreasonable. The fixed threshold has no physical meaning and no consideration of the different states of the snow season. In fact, the snow season is signaled when snowfall can be stabilized into snow on the ground. The currently implemented 10% dynamic threshold method partially addresses some of the

225 limitations of the fixed threshold method, though it is not yet fully perfect. Moreover, the 10% threshold is primarily derived from mathematical calculations. In future work, we plan to integrate atmospheric variables to further enhance the physical significance of the 10% dynamic threshold.

For the second question, the rationalization of dynamic snow phenology methods

230 should start with the problems of existing fixed methods. The limitations of the fixed threshold method are described in detail in Main Comment 1 and will not be repeated here. Please refer to the answer in Main Comment 1 for details.

6.Based on your analysis, I wonder if your approach means that snow phenology

235 metrics will vary depending on the study area used. For example, the snow depth in the Northern-Xinjiang is generally higher, while it is lower in the TP. ---Case1: Only TP data is used to analyze snow phenology metrics for TP. ---Case2: The data both in the TP and Northern-Xinjiang are used to analyze snow phenology metrics for TP. Are the snow phenology metrics in the TP region different between Case 1 and Case 2?

240 Response:

We think the snow phenology of the TP obtained from Case 1 and Case 2 in your

question is the same. Since the threshold is calculated individually for each grid point without applying a regional average, each grid point has its own specific threshold based on its snow curve. Given the differences in snow conditions between the TP and Xinjiang, the extracted thresholds are inherently distinct. The thresholds for grid points on the TP are derived solely from their own snow characteristics and are independent of the snow conditions in Xinjiang. Therefore, including or excluding Xinjiang's snow data does not affect the threshold values for TP grid points, nor does it influence the resulting snow phenology metrics.

**Minor comments:**

1. Change "on" to "in" in line 67.

Response:

We have revised the sentence.

Line 67:

*In fact, snow cover increases with latitude, with thick and stable snow cover at high latitudes and shallow and short-lived snow cover at middle and low latitudes, especially in the TP.*

2. Change "24 hours" to "daily" in line 95

Response:

The sentence has been modified.

Line 95:

*The dataset is a daily product from 1980-2018 with a spatial resolution of 25067.53 meters, and shows a relative deviation within 30%.*

3. Change "the hydrological year" to "a hydrological year" in line 111

Response:

Thank you for your correction, all statement errors in the manuscript have been corrected.

*SCOD is defined as the first day with the first continuous snow cover exceeding five days in a hydrological year, whereas SCED is the last day with the last continuous snow cover exceeding five days.*

275

**4. How do I understand the "snow index" in Line 132? Snow depth? Snow cover days? NDSI?**

Response:

"Snow index" is a pronoun that can denote any of the elements of snow. For better

280 understanding, we've changed it to snow element and given an example in parentheses afterward.

*To investigate the snow phenology in different areas across the globe, we propose a dynamic threshold for snow phenology.*

285 $$Snow_{ratio} = \frac{Snow - Snow_{min}}{Snow_{max} - Snow_{min}} \ , \tag{2}$$

*where $Snow_{max}$ is the annual maximum snow element (e.g., snow cover fraction, snow depth) and $Snow_{min}$ is the annual minimum snow element.*

**5. Line 101-102: "This data has been validated against … less than 5 cm account for**

290 **approximately 65% of all the data" Please add the reference for this statement to support this "5cm - 65%" number pair.**

Response:

Thank you for your advice. We have added the reference for this statement to support this "5cm - 65%" number pair.

295

*This data has been validated against meteorological observations, and absolute errors of less than 5 cm account for approximately 65% of all the data (Che et al., 2008).*

References:

Che, T., Li, X., Jin, R., Armstrong, R., and Zhang, T.: Snow depth derived from passive

300    microwave remote-sensing data in China. Annals of Glaciology, 49, 145–154. https://doi.org/10.3189/172756408787814690, 2008.

6. Line 141: Why do you use "30-day" moving window? Any specific reasons?

Response:

305    Thanks for the comment. The purpose of smoothing is simply to eliminate noise and avoid the effect of chance snowfall timing on the extraction of snow phenology. Snow changes are more stable and reliable on a monthly scale and are not affected by random snowfall.

To argue this point, we analyze the dynamic threshold using percentage of snow depth

310    extracted from the first-order derivatives under different smoothing windows (see Methods) and present the results in a line graph (Fig. R1). The percentage of snow depth threshold stabilizes when the smoothing window reaches approximately 30 days. The curve exhibits a sharp decline for smoothing windows smaller than 30 days, while for values greater than 30 days, it shows minimal fluctuations. Based on this analysis, we

315    select a smoothing window of 30 days. Although minor adjustments to the smoothing window have little impact on the results, it should not be too small, as this increases sensitivity to noise, nor too large, as it may erase some valid information.

[Figure]

320    **Figure R1**. The relationship between the percentage of snow depth (dynamic threshold) and smooth window in (a) the Northern Hemisphere, (b) the Tibetan

Plateau, (c) 30°N–40°N, (d) 40°N–50°N, (e) 50°N–60°N, and (f) 60°N–75°N. The black line is the original line, the black dot is the specific value for each year, and the red line is the trend line.

325

7. Section 2.1: how did you handle the different spatial resolution of these snow products?

Response:

For snow products with different resolutions, we have standardized them to a resolution

330    of 0.25°. We have now commented in the manuscript.

Line 102:

*For snow products with different resolutions, we have standardized them to a resolution of 0.25°.*

335    8. Line 246-247: "Given that the zonal variations … compared to a fixed threshold." It is not clear.

Response:

We apologize for the lack of clarity in our previous explanation. Our intended meaning is that the spatial distribution characteristics of snow are similar to those of vegetation, both exhibiting a latitudinal zonation. However, while vegetation decreases with

340    increasing latitude, snow increases. The key point is that there are clear similarities between the two. In vegetation phenology extraction methods, both fixed threshold and dynamic threshold methods exist. Numerous studies comparing these approaches have consistently shown that the dynamic threshold method is more reasonable. Given the

345    relationship between snow and vegetation, we hypothesize that a dynamic threshold method should also be more accurate than a fixed threshold method for snow phenology extraction.

This point was not clearly conveyed in the original manuscript, and we have now revised it accordingly.

350    Line 251:

*Given this similarity in zonal variation, we propose a dynamic threshold method for extracting snow phenology, inspired by a commonly used approach in vegetation phenology.*

355     9. Are the days in Fig 2 and Table 2 DOYs (Day of Year)? If it is, 66 of SCOD in table 2 means the snow starts in March. Please check the whole paper's figures again.

Response:

Similar to Figure 1, the hydrological year is still represented here. Annotations have been added to the figure notes and tables.

---

## Author Comment (AC4)

**Response to Reviewer**

Dear Dr. Robinson,

5    We are honored by your interest in our work and greatly appreciate your constructive suggestions. Your insights have significantly contributed to improving our work. Following is our response to your question in detail.

**Main comments:**

The authors introduce a method of defining start dates, end dates, and durations of snow cover over Northern Hemisphere lands that differs from fixed definitions for these variables. This reviewer is left with considerable uncertainties as to how the author's dynamic method is any more useful than the fixed method. One major concern has to do with the use of microwave derived snow depth data to "drive" this approach. Many uncertainties remain to this day regarding the accuracy of this spectral region to accurately estimate depth. This can especially be true at shallow depths, with wet snow, and with deep snow that may have ice lenses, depth hoar, etc. within the pack. Unless the authors can explain how these issues do not impact their dynamic phenology method, I would place greater faith in the IMS and SCF estimates. This brings into question just how specific one can get with timing using an 8-day window for the SCF. Not that the IMS is a full-hemisphere daily evaluation of snow cover given persistent cloud cover that may mask surface conditions for multiple days. These issues are mainly more challenging in the fall for all sources, in part due to greater cloudiness then than in the spring melt season. Also, due to shallower depths, potential wet conditions, and ephemeral snow cover that can be found everywhere during seasonal snow onset (not that it doesn't exist throughout the season at lower elevation, low-mid latitude regions. Also, the challenges of using microwave data to map snow cover over the Tibetan Plateau has long been recognized, in part, as the authors suggest, due to its shallow depth (over non-mountainous areas) and ephemeral nature.

I will go no further with this evaluation as I see earlier reviewers have posted excellent comments regarding manuscript specifics, to which I agree. At this point, I will conclude that while the manuscript addresses an interesting topic and employs the key snow cover extent data products at regional to continental scales (IMS and MODIS), I am not convinced that the SD data and the subsequent conclusions built upon it are fully supported.

Response:

Thank you for your question. Our responses to your questions and comments are listed below.

(1) The first question is about how the dynamic threshold method is more useful than the fixed threshold method. In this study, we do not explicitly quantify bias due to the

lack of direct observational data on snow phenology. We also agree that the data have a great impact on the identification of snow phenology. Given the uncertainties in snow data and the absence of snow phenology references, we focus on the limitations of the existing methods and try to improve the extraction of snow phenology in principle.

We all know that snow at high latitudes differs greatly from snow at low latitudes, and snow changes greatly on a time scale as the climatic context changes. The potential implication of using a fixed threshold is to treat all snow as the same, and the choice of a fixed threshold value is too subjective. We think that when snowfall can be steadily converted into snow on the ground, it means the onset of the snow season. It is the inflection point where we look for a change in the state of the snow curve. For example, an inflection point where the snow curve changes from a smooth change to a rapid increase, which means that snow starts to accumulate and the snow season arrives. Therefore, although we do not have observed snow phenology data for validation, the dynamic threshold approach is more reasonable in principle because this approach can take into account the spatial and temporal variability of snow.

(2) The second question is the choice to use snow depth as the driving data for the dynamic thresholding method. As you say, passive microwave remote sensing still faces considerable uncertainties in accurately estimating snow depth.

This technique detects snow cover based on the volume scattering properties of snow particles. The brightness temperature emitted from the surface propagates through the snowpack and undergoes scattering. Uncertainties in passive microwave remote sensing come from a number of sources, such as ground temperature, snow characteristics, and terrain. Under dry snow conditions, the accuracy of passive microwave retrievals improves as snow depth increases. Studies have shown that microwave retrievals tend to underestimate snow extent during autumn and early winter due to the weak scattering signal from shallow and intermittent snow cover. By mid-winter and spring, the error will be relatively small (Armstrong & Brodzik, 2001; Savoie et al., 2009). Notably, the Tibetan Plateau (TP) represents a unique exception within the Northern Hemisphere, where microwave-based retrievals tend to systematically overestimate snow-covered

areas (Frei et al., 2012; Dai et al., 2017). This regional variation highlights the challenges associated with microwave remote sensing of snow, especially in shallow snow areas. On the other hand, in addition to the characteristic of shallowness, you also mention the temporal discontinuity of snow on the Tibetan plateau. Recent studies have shown an average of 14 snow cover events per year in the TP and long periods without snow cover (Li et al., 2022; Wang et al., 2024). All these reasons make TP one of the most difficult areas for passive microwave remote sensing inversion accuracy. Therefore, we will next explore the availability of passive microwave remote sensing snow depth data using TP as the study area.

We use the observational snow depth dataset of the Tibetan Plateau from the National Tibetan Plateau Data Center as a benchmark to validate passive microwave remote sensing snow depth data. There are a total of 102 meteorological stations within the study area, with 99 stations remaining after eliminating three stations with a high number of missing measurements. The time period covers 1961 to 2013, and we select hydrologic years 2001 to 2013 for this study. The hydrological year is defined as the period from September 1 to the following August 31.

To explore the continuity of different snow conditions on the Tibetan Plateau, we select the 30 meteorological stations with the greatest maximum snow depth to represent the deep snow and the 30 stations with the lowest maximum snow depth to represent the shallow snow. The maximum snow depth at deep snow stations is about 12 cm, and the maximum snow depth at shallow snow stations is about 3 cm. Next, we compare the maximum number of consecutive days for shallow snow and deep snow. As we expected, the maximum number of consecutive days for deep snow is longer than for shallow snow. The maximum number of consecutive days for deep snow is centered on 4 days, while for shallow snow, it is only 1 day. Such a short succession of days reflects the instability of snow on the Tibetan Plateau, which increases the difficulty of passive microwave remote sensing inversion.

[Figure]

Figure R1. Spatial distribution of maximum snow depth at 30 meteorological stations of (a) shallow snow and (b) deep snow. (c) The probability distribution function for the maximum number of consecutive days of snow depth at shallow and deep snow weather stations. The blue line represents shallow snow, and the red line represents deep snow.

To compare passive microwave remote sensing data with in situ observations, we select the nearest grid point based on the latitude and longitude of each meteorological station. Spatial maps of multi-year average snow depths show that passive microwave remote sensing snow depths are greater than observed data. Passive microwave remote sensing of snow depth captures the inter-annual variability of the observed data, but suffers from systematic bias and greater fluctuations in observations. The passive microwave remote sensing data do not capture these small fluctuations. It is worth noting that the small number of meteorological stations in TP and the fact that they are mainly located in valleys at lower altitudes and on leeward slopes may make the observations not entirely accurate either.

[Figure]

**Figure R2**. Spatial distribution of average snow depth at 99 sites of (a) observational data, (b) passive microwave remote sensing data. (c)Interannual fluctuations in snow depth for hydrologic years 2001-2013. The solid blue line represents observed snow depth data, and the dashed red line represents passive microwave remote sensing snow depth data.

We further analyze the snow phenology results of passive microwave remote sensing snow depth in comparison with the observed data. First, we use the traditional fixed threshold method for snow phenology extraction. Snow cover duration (SCD) from observed data is generally less than 20 days, with a maximum of 62 days. Passive microwave remote sensing data, because of its large bias, reaches the threshold more often, and the extracted SCD is naturally longer. Twenty-two of these stations have SCDs greater than 70 days. For the snow cover onset day (SCOD) and the snow cover end day (SCED), more stations in the observed data do not extract the SCOD and SCED because it is more difficult to meet the requirement of 5 consecutive days. The SCOD of observational data is generally later than the SCOD of passive microwave remote sensing data, while SCED is the opposite.

[Figure]

**Figure R3**. Spatial distribution of average snow cover duration (SCD), snow cover onset day (SCOD), and snow cover end day (SCED) at 99 sites of observational data (a, c, e) and passive microwave remote sensing data (b, d, f) extracted using the fixed threshold method. The unit DOHY stands for day of the hydrological year.

For comparison with the fixed threshold method, we extract SCD, SCOD, and SCED using our proposed 10% dynamic threshold method, showing that the SCD are prolonged for both data sets. The maximum SCD for observed snow depth data is 77 days. A portion of the sites where SCOD and SCED could not be extracted by the fixed threshold method are also able to obtain snow phenology. The traditional 2cm threshold is too high and difficult to reach for most sites at TP, but that doesn't mean that TP doesn't have a snow season. After using the dynamic approach, the thresholds are selected according to the snow conditions at the TP site itself. Therefore, the dynamic thresholding method will extract more realistic and reasonable snow phenology.

[Figure]

**Figure R4**. Spatial distribution of average snow cover duration (SCD), snow cover onset day (SCOD), and snow cover end day (SCED) at 99 sites of observational data (a, c, e) and passive microwave remote sensing data (b, d, f) extracted using the dynamic threshold method. The unit DOHY stands for day of the hydrological year.

From the interannual variation in snow phenology, the SCD of passive microwave remote sensing of snow depth is longer than the observed data, the SCOD is earlier, and the SCED is later. Overall, passive microwave retrievals tend to depict an extended snow season. Most discrepancies in snow phenology between the two datasets fall within one month. Additionally, the application of the dynamic threshold method results in a longer snow season for both datasets.

[Figure]

**Figure R5.** Interannual fluctuations in snow phenology extracted using (a) the fixed threshold method and (b) the dynamic threshold method for hydrologic years 2001-2013. The solid line represents observed snow depth data, and the dashed line represents passive microwave remote sensing snow depth data. Green represents snow cover duration (SCD), red represents snow cover onset day (SCOD), and yellow represents snow cover end day (SCED).

Snow depths from passive microwave remote sensing in TP are generally higher than observed data, leading to some bias in snow phenology as well. However, a major limitation of optical remote sensing is cloud contamination, which obscures snow data on most days during the snow season. To mitigate this issue, 8-day and 16-day composite snow cover products are generated to reduce cloud interference (Hall et al., 2002). And the resolution of 8 days may be too coarse for extracting snow phenology. The IMS is an unchangeable threshold for processed data. Therefore, each type of data has its own advantages and disadvantages, and we originally chose snow depth as the driving data. In order to make the work more complete, we also use the snow cover fraction as the driving data for the dynamic threshold method of snow phenology. Snow cover yielded roughly the same results as SD, and we show the results in the supporting information. For issues and implications arising from passive microwave remote sensing snow depth data, we have added a description in the discussion section of the manuscript.

In summary, we have recognized the impact of data uncertainty on snow phenology identification, and we believe that the spatial and temporal heterogeneity of snow cover

needs to be considered. This work is a preliminary exploration of dynamic snow phenology, and we hope to further refine it in future work, including your important suggestions.

Line 449:

185 *Since the accuracy of passive microwave detection increases with snow depth, the passive microwave remote sensing data is more effective for analyzing snow phenology in regions with consistent snow cover (Armstrong & Brodzik, 2001; Savoie et al.,2009). In areas with shallow snow with wet snow, the accuracy of passive microwave remote sensing data is reduced, and snow depth indicator may not accurately capture*

190 *accumulation and melting processes. In addition, for the transient snow area, the snow depth curve is more volatile, which makes the assumed single-peak structure untenable. After comprehensive consideration, the snow cover area may be a more reliable indicator in such cases. Therefore, we perform another extraction of dynamic snow phenology using the snow cover data, and the results are similar to SD, but with greater*

195 *differences in TP (see supplement). Regardless of the threshold method, problems with the data itself increase the uncertainty of snow phenology analysis. Therefore, it is necessary to integrate ground observation data with different remote sensing data to form a more comprehensive and accurate snow phenology extraction system.*

**References:**

Armstrong, R. L., & Brodzik, M. J.: Recent Northern Hemisphere snow extent: A comparison of data derived from visible and microwave sensors. Geophysical Research Letters, 28(19), 3673−3676. https://doi.org/10.1029/2000GL012556, 2001.

Dai, L., Che, T., Ding, Y., & Hao, X.: Evaluation of snow cover and snow depth on the Qinghai–Tibetan Plateau derived from passive microwave remote sensing. The Cryosphere, 11(4), 1933–1948. https://doi.org/10.5194/tc-11-1933-2017, 2017.

Frei, A., Tedesco, M., Lee, S., Foster, J., Hall, D. K., Kelly, R., and Robinson, D. A.: A review of global satellite-derived snow products. Advances in Space Research, 50(8), 1007–1029. https://doi.org/10.1016/j.asr.2011.12.021, 2012.

Hall, D. K., Riggs, G. A., Salomonson, V. V., DiGirolamo, N. E., and Bayr, K. J.: MODIS snow-cover products, Remote Sens. Remote Sensing of Environment, 83, 181–194. https://doi.org/10.1016/S0034-4257(02)00095-0, 2002.

Li, H., Zhong, X., Zheng, L., Hao, X., Wang, J., and Zhang, J.: Classification of Snow Cover Persistence across China. Water, 14, 933. https://doi.org/10.3390/w14060933, 2022.

National, M., Tibet, M.: Observational snow depth dataset of the Tibetan Plateau (Version 1.0) (1961-2013). National Tibetan Plateau / Third Pole Environment Data Center. https://doi.org/10.11888/Snow.tpdc.270558, 2018.

Savoie, M. H., Armstrong, R. L., Brodzik, M. J., & Wang, J. R.: Atmospheric corrections for improved satellite passive microwave snow cover retrievals over the Tibet Plateau. Remote Sensing of Environment, 113(12), 2661–2669. https://doi.org/10.1016/j.rse.2009.08.006, 2009.

Wang, J., Tang, L., and Lu, H.: The new indices to describe temporal discontinuity of snow cover on the Qinghai-Tibet Plateau. Npj Climate and Atmospheric Science, 7, 189. https://doi.org/10.1038/s41612-024-00733-y, 2024.

---

## Author Response (AR2)

**Response to Reviewer**

Dear reviewer,

5    We sincerely appreciate your thoughtful comments and detailed suggestions. Your feedback has significantly improved the clarity, rigor, and overall quality of our work. We believe we have adequately addressed all the major and minor comments, and the manuscript has been greatly improved. Our point-by-point responses to your comments are listed below in black.

 The previous reviewers have provided excellent and thorough feedback on the manuscript, with which I largely concur. Upon reviewing your work, I believe your primary contribution to the snow science community lies in proposing a novel snow phenology determining method. The revised manuscript is substantial and well-developed around your proposed algorithm. However, several major issues should be

 addressed to further enhance the manuscript's clarity, rigor, quality, and impact.

**Response:**

Thank you for recognizing our work and for your suggestions. We have taken your comments, and the manuscript has been further enhanced.

 **Main comments:**

1. Regarding research gap (Also noted by Reviewer 2#): Generally, the research gap of one publication should be demonstrated by prior researchers or by author himself. The research gap of your manuscript is currently underdeveloped. Your statement—"The fixed threshold for snow phenology fails to account for the variations in snow cover

 across the NH" and "Snow conditions are variable, but thresholds are always fixed, which can lead to inaccuracies or overestimation of in snow phenology"—reads more like an assumption than a well-supported argument. There is no cited literature, empirical evidence, or demonstration to validate this claim. To strengthen your manuscript, you must either: 1) cite prior studies that highlight limitations of fixed

 thresholds, and/or 2) Provide original analysis (e.g., comparative assessments) to substantiate this assertion. A compelling case for your work's necessity should be made early in the manuscript to convince readers of its significance.

**Response:**

Thank you for your suggestion. We have now revised the manuscript to include

 additional references to highlight the limitation of fixed threshold.

For the threshold of snow depth data, Gascoin et al. (2015) suggested using 10 cm, while other researchers reported best performances for lower values, e.g. 2–5cm (Thirel et al., 2012). Notarnicola (2020) argued that global snow cover analyses should consider

snow characteristics such as accumulation, duration, and melt. The selection of thresholds is optimized according to different snow cover characteristics (in terms of regularity and maximum snow depth). The study showed that in areas where maximum snow depth is below 100 cm, the most suitable threshold ranges from 0 to 10 cm (average 9 cm), while in areas with deeper snow, the optimal range is 5 to 15 cm (average 12.4 cm). In addition, Gascoin et al. (2015) identified a clear seasonal trend in the optimal threshold. These findings suggest that the best threshold varies across space and time, and using a single fixed value may lead to uncertainty in the results.

The following are some auxiliary proofs. Previous studies on vegetation phenology have clearly identified the limitations of fixed-threshold methods (White et al., 1997; Mo et al., 2012). Moreover, the daily snow cover product of the Moderate Resolution Imaging Spectroradiometer (MODIS) relies on the Normalized Snow Cover Index (NDSI) threshold to determine whether there is snow at the grid points. Riggs et al. (2017) suggested that applying a single threshold without accounting for local snow properties, atmospheric conditions, and land cover types often reduces snow detection accuracy. These findings underscore the need for flexible, context-specific threshold approaches in snow cover studies.

Based on the above content, we have revised the manuscript.

Line 58:

*Numerous studies have demonstrated that improving the accuracy of snow cover products is the primary means of enhancing snow phenology metrics (Frei et al., 2012; Estilow et al., 2015; Xiao et al., 2024). However, whether the extraction methods of snow phenology are reasonable has received little attention. Notarnicola (2020) suggested that global snow cover analyses should consider snow characteristics such as accumulation, duration, and melt. The selection of thresholds is optimized according to different snow cover characteristics (in terms of regularity and maximum snow depth). However, most studies still employ a fixed threshold to extract snow phenology in different regions and years (Brown et al., 2007; Gao et al., 2011; Yue et al., 2022; Tang et al., 2022). In fact, applying a single threshold without accounting for local snow*

*properties, atmospheric conditions, and land cover types often reduces snow detection accuracy (Riggs et al., 2017; Gao et al., 2019). Snow cover increases with latitude, with thick and stable snow cover at high latitudes and shallow and short-lived snow cover at middle and low latitudes, especially in the TP (Orsolini et al., 2019). In addition, the snow changes from year to year due to many aspects of the climate and the regional snow cover trends exhibit a heterogeneous and non-linear response to its regional warming rate (Blau et al., 2024). Snow conditions are variable, but thresholds are always fixed, which can lead to uncertainties in snow phenology. At present, it has been proven in the methods of extracting vegetation phenology that fixed thresholds cannot accommodate spatio-temporal heterogeneity, ignore inter-annual variations, and are not applicable to diverse vegetation types, among a series of other problems (White et al., 1997; Mo et al., 2012). We aim to propose a novel method that incorporates both spatial heterogeneity in snow cover and temporal variability to extract snow phenology, reducing the uncertainty associated with the fixed threshold method from a physical meaning perspective.*

2. Regarding the claim: "fixed thresholds always lead to inaccuracies in snow phenology" (also noted by Dr. Xiao). I partially disagree with this statement (or perhaps I have not fully grasped your reasoning). Conventionally, inaccuracies in snow phenology metrics are attributed to errors in snow cover products (e.g., snow cover extent, fraction, depth, or SWE). For example, Dr. Xiao's work(https://doi.org/10.1016/j.isprsjprs.2024.07.018) and other studies focus on improving snow cover product accuracy as a primary means to enhance snow phenology metrics—a point also reflected in your Section 3.1. To support your argument, you must: 1) provide evidence fixed thresholds (independent of snow product errors) contribute significantly to phenology inaccuracies. 2) Acknowledge that snow cover product quality remains a critical source of uncertainty. A broader discussion on all potential uncertainty sources (thresholds, data quality, etc.) would provide a more balanced perspective.

**Response:**

Following the suggestions of Dr. Xiao and your feedback, we have revised the relevant statements. We now place greater emphasis on the limitations of using a fixed threshold. In particular, we highlight that a fixed threshold cannot capture the actual evolution of snow cover across different regions and years.

(1) Regarding the first question, the following are several examples we provided in the manuscript, in which we also expressed our concern about the use of fixed thresholds. As shown in Fig. 4, fixed thresholds are not suitable, especially in mid- and low-latitude regions. For example, in the Tibetan Plateau and the 30°N-40°N zone, SCF rarely reaches 50% throughout the year. For snow depth, the 2 cm mark often corresponds to the middle or even the peak of the snow curve, long after the snow season has begun. We can also find large differences in snow peak values between high and low latitudes, which makes the period when the threshold is reached different. These observations highlight a key limitation: fixed thresholds are limited for extracting snow phenology, regardless of the data type used.

[Figure]

**Figure 4.** Intra-annual variations of IMS, MOD10C2, and SD in five latitudinal zones (including the Tibetan Plateau) from 2000 to 2018. The dashed lines in the MOD10C2 curve represent the MOD10C2 of 50%, and the dashed lines in the SD curves represent the SD of 2 cm. The MOD10C2 dataset north of 60°N is not analyzed due to

the effects of the polar night. The values in the graphs characterize the annual snow maximum over the respective areas.

120    Because the Tibetan Plateau is at low to mid-latitudes while also having areas of stable thick snow and unstable thin snow, we discuss the applicability of dynamic and fixed thresholds to the Tibetan Plateau separately in the manuscript (Fig. 10). Employing the dynamic snow threshold method in mountain areas led to a reduction from the original 2 cm threshold to 0.712 cm, which is more consistent with the position of the turn

125    change in the curve slope. Conversely, snow in non-mountain areas remains shallow and unstable, with an annual average maximum SD below 2.5 cm. The 2 cm threshold nearly reaches the snow peak.

[Figure]

**Figure 10.** Intra-annual variation of snow depth in (a) mountain, and (b) non-
130    mountain areas on the Tibetan Plateau (TP). The red dashed line represents the location of the fixed threshold of 2 cm, and the blue line represents the location of the threshold extracted using the snow dynamic threshold method. (c) Histogram of snow phenology in non-mountain and mountain regions of the TP using the dynamic threshold method. The unit DOHY is an abbreviation for day of the hydrological year,
135    defined as September 1 through August 31 of the following year.

We validate the usability of the remote sensing data in the TP by comparing the passive microwave remote sensing data with in situ observations, and we select the nearest grid points based on the latitude and longitude of each weather station. Spatial maps of multi-year average snow depths show that passive microwave remote sensing snow depths are greater than observed data. Passive microwave remote sensing of snow depth captures the inter-annual variability of the observed data, but suffers from systematic bias. This shows that it is difficult to reach the 2cm threshold even in the case of large raw data, and it is even more difficult to reach the 2cm threshold if the data precision is higher.

[Figure]

**Figure R1**. Spatial distribution of maximum snow depth at 30 meteorological stations of (a) shallow snow and (b) deep snow. (c) The probability distribution function for the maximum number of consecutive days of snow depth at shallow and deep snow weather stations. The blue line represents shallow snow, and the red line represents deep snow.

We plot both absolute and normalized snow depth over the mountainous regions of the Tibetan Plateau for the hydrological years 1988–2017. The timing and magnitude of peak snow depth vary greatly between years, yet the shape of the normalized curves remains consistent. This indicates that the time required to reach a 2 cm snow depth differs from year to year. In years with deeper snow, the 2 cm threshold tends to occur later. By contrast, the 10% dynamic threshold aligns more consistently across all 30 years, typically appearing at the inflection point where the snow line shifts from a

gradual to a rapid rise.

160

[Figure]

**Figure R2**. Intra-annual variations in (a) snow depth and (b) normalized snow depth on the Tibetan Plateau from 1988 to 2017. Gray dashed lines indicate 2cm and 10% respectively. The unit DOHY is an abbreviation for day of the hydrological year, defined as September 1 through August 31 of the following year.

165

These findings show that the uncertainty associated with fixed thresholds cannot be ignored. We have revised the manuscript accordingly.

170   Line 253:

*Based on the above results, we believe that improving the accuracy of snow cover products is one of the primary means for enhancing snow phenology, and that the soundness of snow phenology extraction methods is crucial. In the next section, we aim to enhance the snow phenology extraction method using SD data to obtain more*

175   *reasonable snow phenology in the NH.*

(2) Indeed, we always acknowledge that uncertainties in snow cover products contribute significantly to snow phenology analyses, regardless of the thresholding method used. In Section 3.1 of our manuscript, we show clear differences in snow cover results derived from different products. Therefore, improving the accuracy of snow cover products is the first step to ensure the accuracy of snow phenology results. We also believe this point is well supported and have discussed it in detail in the first part of our discussion (Line 472- Line 487).

In order to provide the reader with a clearer understanding of our point of view, we first add a brief acknowledgement of the impact of snow cover products on the accuracy of snow phenology in the introduction section of the manuscript. Second, a more detailed discussion is provided in the discussion section.

Line 56:

*Various types of snow data are used to extract snow phenology, including SDs, snow cover fractions, and snow water equivalents, leading to possible differences in identified snow phenology (Chen et al., 2015; Guo et al., 2022). Numerous studies have demonstrated that improving the accuracy of snow cover products is the primary means of enhancing snow phenology metrics (Frei et al., 2012; Estilow et al., 2015; Xiao et al., 2024). However, whether the extraction methods of snow phenology are reasonable has received little attention. For example, most studies have employed a fixed threshold to extract snow phenology in different regions and years (Brown et al., 2007; Gao et al., 2011; Yue et al., 2022; Tang et al., 2022).*

Line 472:

*Despite improvements in snow phenology extraction, variations in the data and definitions of snow phenology and hydrological years lead to differences in extracted snow phenological characteristics, which are further compounded by inherent data uncertainties (Xie et al., 2017; Ma et al., 2020; Guo et al., 2022). The fundamental principles underlying snow information acquisition vary across observation methods, impacting binary snow results (Hall and Riggs, 2007; Dietz et al., 2011; Zhang et al., 2024). Factors like the observational instrument's orbit and cloud cover can further*

*affect the accuracy of snow datasets (2005; Gao et al., 2010; Coll and Li, 2018). Second, the performance of snow data varies geographically. Since the accuracy of passive microwave detection increases with snow depth, the passive microwave remote sensing data is more effective for analyzing snow phenology in regions with consistent snow cover (Armstrong & Brodzik, 2001; Savoie et al.,2009). In areas with shallow snow with wet snow, the accuracy of passive microwave remote sensing data is reduced and the snow depth indicator may not accurately capture accumulation and melting processes. In addition, for the transient snow area, the snow depth curve is more volatile, which makes the assumed single-peak structure untenable. After comprehensive consideration, the snow cover fraction may be a more reliable indicator in such cases. Therefore, we perform another extraction of dynamic snow phenology using the snow cover fraction data, and the results are similar to SD, but with greater differences in TP (see Fig. S1- S5 in Supplement). Regardless of the threshold method, problems with the data itself increase the uncertainty of snow phenology analysis. Therefore, it is necessary to integrate ground observation data with different remote sensing data to form a more comprehensive and accurate snow phenology extraction system.*

3. Some suggestions on logical flow: The manuscript appears to follow a straightforward logic: propose a new algorithm and validate its performance through extensive analysis. However, as you noted in your responses, ground-truth observations for snow phenology metrics are lacking, making absolute accuracy assessments impossible. Thus, claims of your method's "accuracy" should be tempered. Instead, focus on demonstrating its reasonableness (e.g., via consistency checks, comparative advantages, or physical plausibility). I recommend revising statements throughout the manuscript to reflect this nuance.

**Response:**

We appreciate the reviewer's thoughtful comments. After receiving suggestions from previous reviewers and community members, we have also recognized that the absence of observational data on snow phenology prevents us from validating which method is

235 absolutely more accurate. In addition, our aim is to improve on traditional snow phenology methods, not to discredit them. In response, we revise our terminology. While we propose a dynamic threshold method grounded in physical processes, we now refer to it as more reasonable, rather than more accurate. Compared with the traditional fixed threshold approach, the dynamic method offers clearer physical relevance and

240 greater flexibility. Accordingly, we have replaced all instances of "accurate" with "reasonable" when describing the dynamic identification of snow phenology in the manuscript. We have also replaced "errors" and "bias" in the fixed threshold method with "limitations".

Line 7:

245 *Previous studies commonly employed fixed threshold methods to extract snow phenology, which cannot represent the differences in the beginning/end of the snow period under different snow conditions in the Northern Hemisphere, leading to potential uncertainties in snow phenology.*

Line 70:

250 *We aim to propose a novel method that incorporates both spatial heterogeneity in snow cover and temporal variability to extract snow phenology, reducing the uncertainty associated with the fixed threshold method from a physical meaning perspective.*

Line 254:

*In the next section, we aim to enhance the snow phenology extraction method using SD*
255 *data to obtain more reasonable snow phenology in the NH.*

Line 314:

*This suggests that the method can dynamically adjust the threshold based on the annual and regional SD's variations, thereby reducing the uncertainty in snow phenology extraction caused by large-scale climatic influences.*

260 Line 377:

*In addition, the snow phenology dynamic threshold method is more reasonable in areas with complex topographic and climatic features, such as the TP.*

**Minor comments:**

265  1.  Table1: "The four days" ---> "The fourth day"

The definitions in Table 1 seem tailored to your called "fixed threshold". If so, clarify this in the table title. If not, explain how your proposed algorithm accommodates these definitions across different snow cover products in your updated manuscript.

**Response:**

270  Thank you for your correction, and the sentence and title has been modified.

Table 1:

*the fourth day before which the pixel is first covered with snow.*

Line 117:

**Table 1.** *Definitions of fixed threshold methods for snow phenology parameters with*

275  *different datasets.*

2. Equation 2: The range of snow cover fraction is from 0 to 1 over the whole seasonal snow season. The equation includes "snow cover fraction", but the variable "Snow_ratio" appears equivalent to "Snow (snow cover fraction)". This raises

280  questions about the equations applicability, especially given your note that it cannot be used with IMS data. Clarify how your algorithm derives phenology metrics in such cases.

**Response:**

Although both $Snow_{ratio}$ and snow cover fraction (SCF) range from 0 to 1, they represent

285  different physical meanings. The SCF is spatial dimension and represents the distribution state of snow at a certain moment. In contrast, $Snow_{ratio}$ captures a key transition point over time. When the snow depth is input as snow element, if the maximum snow depth is 20cm (the minimum snow depth is 0), the 10% threshold corresponding to it is 2cm. When SCF is used as an element, if the maximum SCF is

290  50% (the minimum SCF is 0), its 10% threshold corresponds to 5%. $Snow_{ratio}$ represents the key turning point of the change in snow coverage in a grid/area, corresponding to a certain snow element value. Once this threshold is reached, we consider the grid to be snow-covered.

We do not apply this formula to IMS data because it has already been processed. The IMS product classifies each grid cell as snow-covered or snow-free based on a set threshold. As a result, we do not reassess snow presence or compute grid-level percentage changes at the grid level.

3. Fig. 7: "percentage of snow depth"--->"Normalized Snow depth". The current term suggests a direct accumulation of values, which may mislead readers.

**Response:**

Thank you for your advice. To avoid misunderstanding, we have changed "percentage of snow depth" to "Snow depth Normalized Radio" in the figure.

4. One friendly suggestion: While borrowing concepts from vegetation phenology is innovative, fundamental differences between snow and vegetation dynamics (e.g., snow's rapid appearance/disappearance vs. gradual vegetation changes) require careful handling. Explicitly address these discrepancies and justify any adapted methodologies.

**Response:**

Thank you for the suggestion. As you point out, snow and vegetation are similar in some ways, but fundamentally different. We use the analogy between vegetation and snow to highlight a broader point: diverse extraction methods are also needed for snow phenology. Vegetation phenology research began earlier and is now relatively mature. Many studies have proposed various methods to extract vegetation phenology (Zhang et al., 2003; Filippa et al., 2016; Kong et al., 2022). In contrast, snow phenology has typically relied on a single method with a fixed threshold. We believe this limits progress and innovation in the field. The dynamic threshold method builds on ideas from vegetation phenology but does not assume that snow and vegetation behave in the same way. For instance, vegetation studies often use a 15% threshold. In our work, we identify a 10% dynamic threshold for snow depth by considering both first-order derivatives and the underlying physical processes. Our research results show that the 10% threshold is applicable in most areas of the Northern Hemisphere. Moreover,

because snow changes rapidly, we require more than just reaching a threshold to define snow phenology. We also apply a five-day continuity condition to ensure stability. This second step is crucial—it clearly distinguishes snow phenology from vegetation phenology. We will further clarify the key differences between snow and vegetation, while emphasizing the need for more flexible and physical methods in snow phenology research. Based on the above, we add the following sentences in the discussion section.

Line 488:

*We propose a dynamic approach to defining snow phenology by adjusting the threshold for snow presence in this study. However, snow and vegetation differ in their fundamental dynamics. Vegetation grows gradually, while snow can change rapidly over short periods. Therefore, there is a second key step in the extraction of snow phenology, which requires that the threshold must be met for several consecutive days. This condition ensures that the detected event reflects a stable and meaningful snow presence and mitigates the influence of sporadic snowfall events.*

**References:**

Filippa, G., Cremonese, E., Migliavacca, M., Galvagno, M., Forkel, M., Wingate, L.,
Tomelleri, E., Morra di Cella, U., & Richardson, A. D.: Phenopix: A R package for
image-based vegetation phenology. Agricultural and Forest Meteorology, 220,
141–150. https://doi.org/10.1016/j.agrformet.2016.01.006, 2016.

Gascoin, S., Hagolle, O., Huc, M., Jarlan, L., Dejoux, J.-F., Szczypta, C., Marti, R., &
Sánchez, R.: A snow cover climatology for the Pyrenees from MODIS snow
products. Hydrology and Earth System Sciences, 19(5), 2337–2351.
https://doi.org/10.5194/hess-19-2337-2015, 2015.

Kong, D., McVicar, T. R., Xiao, M., Zhang, Y., Peña-Arancibia, J. L., Filippa, G., Xie,
Y., & Gu, X.: phenofit: An R package for extracting vegetation phenology from
time series remote sensing. Methods in Ecology and Evolution, 13(7), 1508–1527.
https://doi.org/10.1111/2041-210X.13870, 2022.

Mo, J., Zhu, W., Wang, L., Xu, Y., and Liu J.: Evaluation of remote sensing extraction
methods for vegetation phenology based on flux tower net ecosystem carbon
exchange data. Chinese Journal of Applied Ecology, 23(2), 319-327.
https://doi.org/10.13287/j.1001-9332.2012.0072, 2012.

Notarnicola, C. : Hotspots of snow cover changes in global mountain regions over
2000–2018. Remote Sensing of Environment, 243, 111781.
https://doi.org/10.1016/j.rse.2020.111781, 2020.

Riggs, G.A., Hall, D.K., Salomonson, V.V.: MODIS snow products user guide to
collection 5. Digital Media 80, 1–80, 2006.

Riggs, G. A., Hall, D. K., & Román, M. O.: Overview of NASA's MODIS and Visible
Infrared Imaging Radiometer Suite (VIIRS) snow-cover Earth System Data
Records. Earth System Science Data, 9, 765-777. https://doi.org/10.5194/essd-9-765-2017, 2017.

Thirel, G., Notarnicola, C., Kalas, M., Zebisch, M., Schellenberger, T., Tetzlaff, A.,
Duguay, M., Mölg, N., Burek, P., & De Roo, A. : Assessing the quality of a real-time Snow Cover Area product for hydrological applications. Remote Sensing of
Environment, 127, 271–287. https://doi.org/10.1016/j.rse.2012.09.006, 2012.

White, M. A., Thornton, P. E., and Running, S. W.: A continental phenology model for monitoring vegetation responses to interannual climatic variability. Global Biogeochemical Cycles, 11, 217–234. https://doi.org/10.1029/97GB00330, 1997.

Zhang, X., Friedl, M. A., Schaaf, C. B., Strahler, A. H., Hodges, J. C. F. F., Gao, F., Reed, B. C., & Huete, A.: Monitoring vegetation phenology using MODIS. Remote Sensing of Environment, 84(3), 471–475. https://doi.org/10.1016/S0034-4257(02)00135-9, 2003.

370